# SUMOylation of Jun fine-tunes the *Drosophila* gut immune response

**Amarendranath Soory** *, **Girish S. Ratnaparkhi** [ID] *

Department of Biology, Indian Institute of Science Education & Research, Pune, India

* amarendranath.soory@students.iiserpune.ac.in (AS); girish@iiserpune.ac.in (GR)

## Abstract

Post-translational modification by the small ubiquitin-like modifier, SUMO can modulate the activity of its conjugated proteins in a plethora of cellular contexts. The effect of SUMO conjugation of proteins during an immune response is poorly understood in *Drosophila*. We have previously identified that the transcription factor Jra, the *Drosophila* Jun ortholog and a member of the AP-1 complex is one such SUMO target. Here, we find that Jra is a regulator of the *Pseudomonas entomophila* induced gut immune gene regulatory network, modulating the expression of a few thousand genes, as measured by quantitative RNA sequencing. Decrease in *Jra* in gut enterocytes is protective, suggesting that reduction of Jra signaling favors the host over the pathogen. In Jra, lysines 29 and 190 are SUMO conjugation targets, with the $Jra^{K29R+K190R}$ double mutant being SUMO conjugation resistant (SCR). Interestingly, a $Jra^{SCR}$ fly line, generated by CRISPR/Cas9 based genome editing, is more sensitive to infection, with adults showing a weakened host response and increased proliferation of *Pseudomonas*. Transcriptome analysis of the guts of $Jra^{SCR}$ and $Jra^{WT}$ flies suggests that lack of SUMOylation of Jra significantly changes core elements of the immune gene regulatory network, which include antimicrobial agents, secreted ligands, feedback regulators, and transcription factors. Mechanistically, SUMOylation attenuates Jra activity, with the TFs, *forkhead*, *anterior open*, *activating transcription factor 3* and the master immune regulator Relish being important transcriptional targets. Our study implicates Jra as a major immune regulator, with dynamic SUMO conjugation/deconjugation of Jra modulating the kinetics of the gut immune response.

## Author summary

The intestine has a resident population of commensal microorganisms against which the immune machinery is tuned to show low or no reactivity. In contrast, when pathogenic microorganisms are ingested, the gut responds by activating signaling cascades that lead to the killing and clearance of the pathogen. In this study, we examine the role played by the well-known transcription factor Jun in regulating the immune response in the *Drosophila* gut. We find that loss of Jun leads to the change in intensity and kinetics of the gut immune transcriptome. The transcriptional profile indicates a stronger response when Jun activity is reduced. Also, animals infected with *Pseudomonas entomophila* live longer

**Data Availability Statement:** The authors confirm that all data underlying the findings are available without restriction. The RNA sequencing data have been deposited in the Gene-Expression Omnibus database (https://www.ncbi.nlm.nih.gov/geo/) as GSE194168 with immediate access. The deposition

contains forty-two datasets with IDs GSM5829976-GSM5830017'.

**Funding:** GSR is supported by Genome Engineering Technology (GET) grant Department of Biotechnology – Genome Engineering technology grant (BT/PR26095/GET/119/199/2017), Department of Biotechnology (DBT), Govt. of India (https://dbtindia.gov.in/) and Scheme for Transformational and Advanced Research (STARS), Ministry of Education grant 730-2019 (https://stars.iisc.ac.in/). The IISER Drosophila media and Stock centre is supported by National Facility for Gene Function in Health and Disease (NFGFHD) at IISER Pune, which in turn is supported by an infrastructure grant (BT/INF/22/SP17358/2016) from the Department of Biotechnology (DBT), Govt. of India (https://dbtindia.gov.in/). AS was supported by: IISER Pune graduate student fellowship and Council of Scientific and Industrial Research (CSIR) – direct SRF fellowship (https://csirhrdg.res.in/). The funders had no role in study design, data collection and analysis, decision to publish, or preparation of the manuscript.

**Competing interests:** The authors have declared that no competing interests exist.

when Jun signaling is reduced. Further, we find that Jun is post-translationally modified on Lys29 and Lys190 by SUMO. To understand the effect of SUMO-conjugation of Jun, we create by state-of-the-art CRISPR/Cas9 genome editing a *Drosophila* line where Jun is resistant to SUMOylation. This line is more sensitive to infection, with a weaker host-defense response. Our data suggest that Jun Signaling favors the pathogen by dampening the immune response. SUMO conjugation of Jun reverses the dampening and strengthens the immune response in favor of the host. Dynamic SUMOylation of Jun thus fine-tunes the gut immune response to pathogens.

## Introduction

Proteins undergo post-translational modifications (PTM) by reversible and covalent attachment of a diverse set of molecules. PTMs can modulate the target protein's structure, stability, interactions, or location and alter its function. This modulation increases the complexity and diversity of a proteome. The effect of the PTM on the target protein is usually context-dependent, with the combination of different PTMs affecting function in a differential manner. PTMs come in all shapes and sizes with a wide variety of chemical groups. One common PTM is the conjugation of one protein to another. The best-known examples of such protein modifiers are Ubiquitin-like proteins (UBLs) [1–4], a large group of proteins exhibiting a Ubiquitin fold. These, along with Ubiquitin, include Nedd8, ISG15, SUMO, ATG8/12, and many others. These protein PTMs conjugate many proteins, usually targeting lysine side chains.

The small ubiquitin-like modifier, SUMO [5–7] can be reversibly and covalently conjugated to a target protein. For conjugation, the translated SUMO protein is made conjugation-competent by proteolytic cleavage of the C-terminal tail of SUMO, exposing a–GG dipeptide. The SUMO-GG is then picked up by the SUMO activating enzymes (SAEs) which transfer the SUMO to the SUMO conjugase Ubc9. Ubc9 interacts with the target protein and works to attach the C-terminal GG-COO- to a lysine side chain, on the target protein, via an isopeptide bond [8,9]. The conjugation is reversible and requires deconjugating enzymes [10]. Ubc9, during conjugation, can be assisted by one or multiple SUMO ligases, which are believed to provide specificity and enhance conjugation.

Our laboratory is interested in the role of SUMO conjugation of proteins during an innate immune response in flies [11,12]. A subset of proteins involved in immune regulatory networks are known to be SUMOylated both in vertebrates or invertebrates [11,13,14]. Our approach to the problem was first to identify proteins that changed their SUMOylation status on infection. We used quantitative proteomics to identify a set of proteins whose SUMOylation state either increased or decreased with infection [12]. One SUMO target uncovered in our quantitative proteomics screen [12] is the *Drosophila* c-Jun ortholog Jra (Flybase ID, FBgn0001291)—the Jun related antigen [15,16]. Jra is one of the transcriptional effectors of the JNK (Jun N-terminal kinase) signaling pathway, the other being Kayak (Kay, ortholog of c-Fos) [17,18]. Jra function has been uncovered in multiple developmental contexts [19–22], as well as in the adult animal [23,24]. The JNK kinase cascade has been broadly classified as a 'stress' pathway in organisms, and our current research focuses on its role in immune signaling.

In this study, we have explored roles for SUMO conjugation of Jra in the gut immune response. Using the loss of function genetics, in combination with quantitative RNA sequencing, we find that Jra is a major regulator of the genes involved in the gut immune response, triggered by oral *Pseudomonas entomophila* (*Pe*) infection. Our data suggest that Jra is part of

an extensive immune Gene regulatory network (i-GRN) and is crucial for evoking a robust immune response and maintaining gut immune homeostasis. Using an antibody against full-length Jra, we also show that Jra abundantly binds to the promoters of *forkhead*, (*fkh*), *anterior open (aop)*, *activating transcription factor 3* (*atf3*) and *rel*. We find that SUMO conjugates Jra at lysines 29 and 190, and a Jra$^{K29R+190R}$ mutant is SUMO-conjugation resistant (SCR). The *Jra$^{SCR}$* double mutant line, generated by CRISPR Cas9 genome editing, is sensitive to *Pe* infection and carries a higher load of bacteria, suggesting that the absence of Jra SUMOylation weakens the ability of the host to resist infection. At the level of transcription, lack of Jra SUMOylation changes the kinetics of the immune response, delaying and weakening the activation of defense genes, an important example being the NFκB factor Relish (Rel).

Our study thus uncovers Jra as a major transcriptional regulator of the gut immune response. Increased Jra signaling appears to favor *Pseudomonas*, while SUMOylation of Jra favors the host, presumably by enhancing the host response to pathogens. Thus, we postulate that SUMOylation of Jra fine-tunes the host transcriptional response to infection.

## Results

### Jra is a target for SUMO conjugation

We initiated our studies by validating our initial observation [12] that Jra is SUMO conjugated (SUMOylated). Jra SUMOylation was validated using a previously described *in-bacto* approach [25]. Here, Jra was co-expressed in bacteria with *Drosophila* SAE1, SAE2, Ubc9, and mature 6XHis-tagged SUMO (SUMO-GG) to facilitate target SUMOylation. Affinity purification using glutathione beads, followed by western blotting, was used to detect SUMOylated species migrating at a higher molecular weight than Jra, based on the degree of conjugation. A similar experiment with an immature/dead 6XHis-tagged SUMO (SUMO-ΔGG) served as a negative control. We consistently find two high molecular weight bands that correspond to 1X and 2X SUMO conjugated species in the Western blots (Fig 1A), confirming that Jra is a target of SUMO conjugation. We also find a third higher molecular weight band corresponding to >3X SUMO conjugated species. GST itself does not get SUMOylated under *in-bacto* experimental conditions (S1 Fig and [25]). To discover the lysine residues within Jra that are targets of SUMO machinery, we used the web-based tool, Joined Advanced SUMOylation Site and Sim Analyser (JASSA) [26] to predict the putative SUMO sites (Fig 1B). SUMO is known to prefer lysine residues that follow the broad consensus SUMO motif (ψ**K**XE/D), where ψ is a hydrophobic residue. A total of seven sites (Fig 1B) were targeted based both on JASSA predictions and the evolutionary conservation of SUMO sites. Of these, K214 was a strong consensus motif while K29 and K190 were part of Negatively charged amino acid-dependent SUMOylation (NSDM) site and Synergy Control (SC)–SUMO motifs, respectively (Fig 1C). The three-dimensional structures of Jun available in the Protein Data Bank do not include the N-terminal and C-terminal regions, with secondary structure predictions [27] suggesting that K29 and K190 are in unstructured regions. We replaced each of the predicted lysine residues, one at a time, with an arginine residue, whose side chain is resistant to SUMO conjugation (Fig 1D). Our goal was to identify SUMO conjugation sites and generate a SUMO conjugation resistant mutant (SCR) of Jra. Of the seven site-directed mutants tested (Fig 1C and 1D), only two, K29R and K190R, showed partial loss of SUMOylation. There were changes in the intensity of SUMOylated bands for the remaining single mutations, but their SUMOylation status was maintained (Fig 1D). As a further step to generate a Jra SCR mutant, we cloned and tested the Jra double mutant, Jra$^{K29R+K190R}$. Interestingly, we found that the double mutant could not be SUMO conjugated, with all three SUMOylation bands missing (Fig 1E). The SUMO sites we have uncovered are distinct from those SUMOylated in c-Jun [28,29], with K29 in flies not

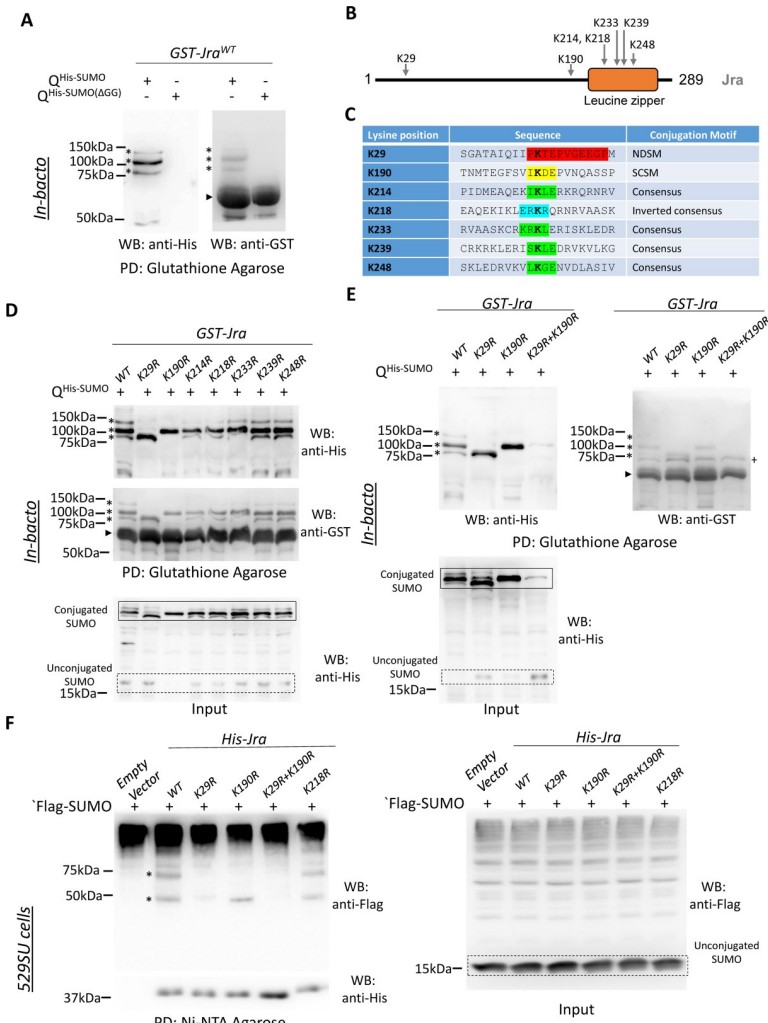

**Fig 1. Site-directed lysine mutagenesis screen identifies Jra(K29) and Jra(K190) as acceptor sites for SUMOylation,. A.** Anti-His western blot (left panel) post Glutathione agarose affinity pulldown (PD) shows three SUMOylated bands for GST-Jra (marked by asterisks, *). *GST-Jra^WT* co-transformed with immature *SUMO (SUMO-ΔGG)* serves as a negative control for SUMO conjugation. Anti-GST WB (right panel) post affinity PD indicates similar levels of GST-Jra^WT. The SUMO conjugated species of Jra are seen as three faint bands (marked by asterisks) above unconjugated Jra (marked by an arrowhead). **B.** Schematic of the primary sequence of *Drosophila* Jra. The conjugation sites to be tested (panel C) are mapped onto Jra. Secondary structure prediction indicates that the Jra N-terminus is unstructured (black line), while the C-terminus (residues 210–275) is helical. This agrees with structures deposited in the Protein Data Bank (PDB:1FOS), which correspond only to the C-terminal leucine zipper (orange box), formed from c-Jun/c-Fos dimers. **C.** Putative SUMO motifs predicted for Jra include the negative amino-acid dependant SUMO motif (NDSM; K29), synergy consensus SUMO motif (SCSM; K190), four consensus SUMO motifs (K214, K233, K239 and K248) and an inverted consensus motif (K218). **D.** Anti-His western blot (upper panel) post PD shows the SUMOylated bands of Jra and its mutant variants depicted by asterisks. Jra^K29R shows partial loss of SUMOylation where the two bands on the top disappear. Also, Jra^K190R shows partial loss of SUMOylation where the upper and lowermost SUMOylated bands disappear. Anti-GST western blot (middle panel) post PD showing different variants of unconjugated GST-Jra (marked by an arrowhead) along with GST-Jra conjugated with SUMO (marked by asterisks). Anti-His WB of the lysates prior to pulldown in the lower panel shows the conjugated and unconjugated SUMO protein. **E.** Anti-His western blot (top left panel) of Jra post PD showing the three distinct SUMOylated Jra bands in Jra^WT. Partial loss of SUMOylation in Jra^K29R and Jra^K190R, as seen in panels D is confirmed. Jra^K29R+K190R shows complete loss of SUMOylation. Anti-GST western blot (top right panel) post PD showing unmodified Jra (marked by an arrowhead) along with modified Jra (marked by asterisks). Anti-His WB of the lysates prior to pulldown in the lower left panel shows the conjugated and unconjugated SUMO protein. **F.** *In-vivo* demonstration of SUMOylation of Jra. Anti-Flag western blot post-Ni-NTA affinity pull-down in the upper left panel represents the SUMOylated species of Jra seen as two distinct bands, marked by asterisks. Jra^K190R shows loss of the upper band, indicating partial loss of SUMOylation. Jra^K29R and Jra^K29R+K190R show complete loss of SUMOylation. Jra^K218R that

was previously shown not to alter SUMOylation status in the bacterial assay, is at par with $Jra^{WT}$. The empty vector was used as a master negative control. Anti-His western blot (lower panel) represents unmodified Jra. The anti-His WB prior to the PD in the right panel serves as input control for expression of SUMO.

conserved in mammals. Furthermore, these sites are not known targets of any other PTM, with K112 being the only known acetylated lysine [30,31] in *Drosophila*.

Based on the molecular weight of the SUMOylated species in the $Jra^{K29R}$ and $Jra^{K190R}$ lanes, we suspect that there is a third SUMO site in Jra which is dependent on SUMOylation of $Jra^{K29}$. However, this third site is not populated either in the $Jra^{K29R}$ single mutant or the $Jra^{K29R+K190R}$ double mutant. Another possibility, di-SUMOylation at $Jra^{K29R}$, exists, but there is evidence against poly-SUMOylation in insects [32].

The $Jra^{K29R+K190R}$ is our putative SCR mutant ($Jra^{SCR}$). To confirm the resistance of this mutant to SUMO conjugation *in-vivo*, we cloned $Jra^{WT}$, $Jra^{K29R}$, $Jra^{K190R}$ and $Jra^{K29R+K190R}$ with a 6XHis tag into an inducible pRM vector and transfected 529SU cells transiently with these constructs. 529SU is a stable S2 cell line [33] that expresses FLAG-SUMO-GG, SAE1, SAE2 and Ubc9 using inducible metallothionein promoters. Expression of Jra variants was initiated by adding CuSO4, which activates the metallothionein promoter. We have performed Ni-NTA affinity pulldown under denaturing conditions, followed by western blots probed with an anti-FLAG antibody and an anti-6XHis antibody (Fig 1F). In S2 cells, $Jra^{WT}$ showed two SUMO-conjugated bands, corresponding to 1X and 2X SUMOylated species. $Jra^{K190R}$ retained one SUMO conjugation site, but interestingly both $Jra^{K29R}$ and $Jra^{K29R+K190R}$ did not show any SUMO conjugation. Since all further experiments were to be conducted in the animal, and both these sites were high confidence SUMO conjugation sites, we chose to work further with the double mutant, $Jra^{K29R+K190R}$, and defined it as $Jra^{SCR}$.

## Jra and SUMO play critical roles in host defense in the fly gut

After exploring published roles for Jra, we decided to explore Jra function in the gut, which is one of the primary immune organs in the fly [34,35]. Jra and the other components of the JNK pathway are expressed in various cell types of the gut, with *Jra* upregulated in response to *Pe* infection [36] (S2 Fig). *Jra* transcripts are highest in the enterocytes (ECs, 4.3-fold), followed by the entero-endocrine cells (EEs, 2.7-fold), the interstitial stem cells (ISCs, 1.6-fold) and the enteroblasts (EBs, 1.2-fold) post-*Pe* infection (S2 Fig). Flies orally fed with *Pe* rapidly succumb despite the induction of both local and systemic immune responses [37,38]. Hence, we used the *Pe* infection model to study the function of Jra in gut immunity. Since Jra null animals are embryonic lethal, we tested the ability of two well characterised null alleles of *Jra*, $Jra^{IA109}$ and $Jra^{76-19}$, to resist *Pe* infection in a heterozygous combination with $w^{1118}$ (Fig 2A). Interestingly, the absence of one copy of *Jra* protected the animal against infection, significantly extending lifespan (Fig 2A). In order to validate this result, we used a gut EC driver, $Myo31DF\text{-}Gal4^{ts}$ (also called $NP1\text{-}Gal4^{ts}$), to reduce Jra function by both RNA interference and by overexpressing a dominant-negative ($Jra^{DN}$) allele. In both experiments (Fig 2B), reduction of Jra activity in the gut led to partial protection against the pathogen, leading to an extension of lifespan.

In contrast, reduction of Jra activity either in the whole animal (Fig 2A) or in the gut (Fig 2B) in the absence of infection does not appear to affect lifespan, suggesting that Jra is haplo-sufficient in this context. In flies, the Jra function is activated by a Jun kinase (JNK), Basket (Bsk) [16,39]. Bsk modulates stem cell proliferation and impacts gut regeneration in *Drosophila* [40–42]. Also, Bsk synergizes with the Imd pathway to regulate epithelial cell shedding during gut infection [43]. However, the role of Bsk in the regulation of the gut immune response is not well understood. To test this, Bsk function was attenuated by overexpressing a

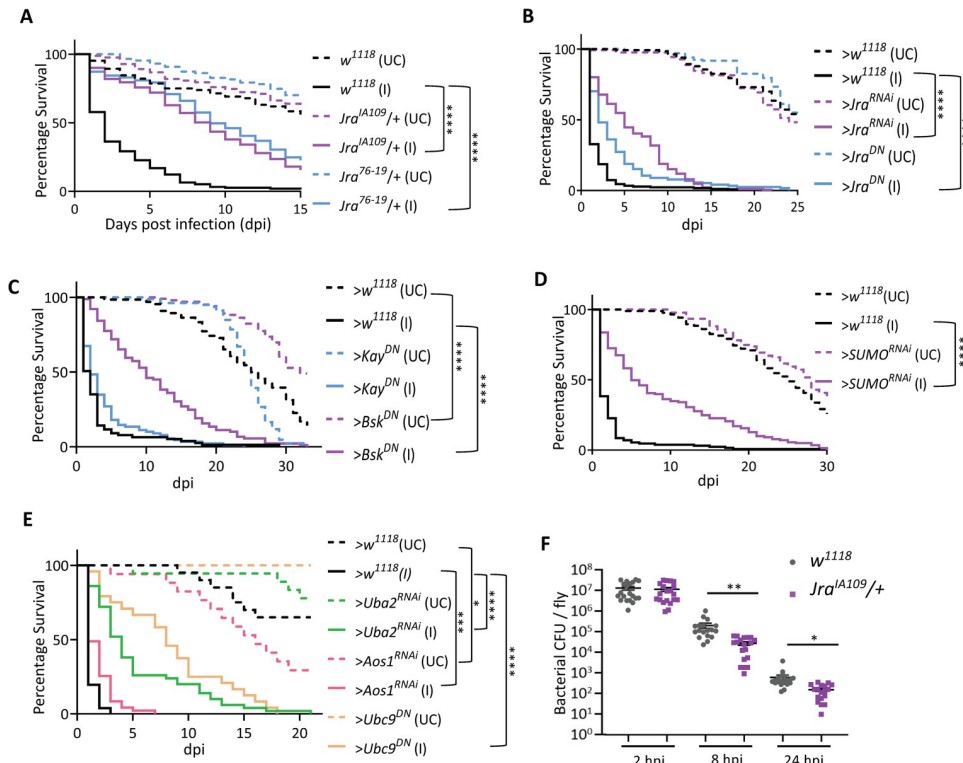

**Fig 2. Jra signalling and components of SUMO conjugation machinery have important host-defence roles in the _Drosophila_ gut. A.** Survival curves of unchallenged (UC, dotted) and infected (I, closed) flies: $w^{1118}$ (black), $Jra^{IA109}$/+ (purple) and $Jra^{76-19}$/+ (blue). Both $Jra^{IA109}$/+ and $Jra^{76-19}$/+ flies show enhanced survival upon infection. Log-rank test for trend was used to compare $Jra^{IA109}$/+ (I) to $w^{1118}$ (I) and $Jra^{76-19}$/+ (I) to $w^{1118}$ (I). ****p<0.0001. **B.** Survival curves of UC (dotted) and I (closed), flies after over-expression of $Jra^{RNAi}$ (purple) and $Jra^{DN}$ (blue) The _Gal4_ control is represented in black. Loss of function of Jra showed enhanced survival upon infection and suggested a gut-specific role of Jra as a suppressor of the immune response. Log-rank test for trend was used to compare >$Jra^{RNAi}$ (I) to >$w^{1118}$ and >$Jra^{DN}$ (I) to >$w^{1118}$ (I). ****p<0.0001. **C.** Survival curves of UC (dotted) and I (closed) flies, post-over-expression of $Bsk^{DN}$ (purple) and $Kay^{DN}$ (blue). The _Gal4_ control is represented in black. Loss of function of _Bsk_ (but not _Kay_) in the gut showed enhanced survival upon infection. Log-rank test for trend was used to compare >$Bsk^{DN}$ (I) to >$w^{1118}$ (I), >$Kay^{DN}$ (I) to >$w^{1118}$ (I) and >$Bsk^{DN}$ (UC) to >$w^{1118}$ (UC). ****p<0.0001. **D.** Survival curves of unchallenged UC (dotted) and I (closed) flies: >$w^{1118}$ (black) and >$SUMO^{RNAi}$ (purple). Loss of SUMO in the gut enhances lifespan post-infection. Log-rank test for trend was used to compare >$SUMO^{RNAi}$ (I) to >$w^{1118}$ (I). ****p<0.0001. **E.** Survival curves of UC (dotted) and I (closed) flies after perturbing different components of the SUMO cycle machinery: >$w^{1118}$ (black), >$Uba2^{RNAi}$ (green), >$Aos1^{RNAi}$ (pink) and >$Ubc9^{DN}$ (orange). Log rank test for trend was used to compare >$Uba2^{RNAi}$ (I) to >$w^{1118}$ (I), >$Aos1^{RNAi}$ (I) to >$w^{1118}$ (I) and >$Ubc9^{DN}$ (I) to >$w^{1118}$ (I) and >$Aos^{RNAi}$ (UC) to > $w^{1118}$ (UC). *p = 0.03; ***p = 0.0008 ****p<0.0001. **F.** Scatter dot plot representing bacterial load as colony-forming units (CFUs) post-infection. The student's t-test was performed comparing $w^{1118}$ and $Jra^{IA109}$/+ at respective time points. **p = 0.0025; *p = 0.0194. _Enterocyte specific NP1-Gal4$^{ts}$ was used to express the transgenes. Data related to the number of animals per experiment and replicates listed in S2A Fig. Data not significant, not represented._

dominant-negative allele ($Bsk^{DN}$) in the ECs, and the flies were infected with _Pe_ (Fig 2C). Interestingly, as in the case of _Jra_ knockdown, reduction of Bsk activity also increased lifespan to a similar extent, suggesting that Bsk functions to regulate Jra activation in the host response to infection (Fig 2C).

Surprisingly, loss of function of Bsk in the gut enhanced lifespan in non-infected conditions. This may suggest that Bsk is a kinase for additional proteins in the gut that regulate the fly's lifespan. Kayak (Kay) can also partner with Jra for transcriptional regulation, with the heterodimeric Jra/Kay complex orthologous to the Jun/Fos AP-1 complex [44,45]. In the fly gut, however, reducing the function of Kay did not appreciably change the lifespan of the adult fly

(Fig 2C) post-infection, possibly indicating a Kay-independent role for Jra in responding to *Pe* infection in the gut.

Since this study aims to understand the roles of Jra SUMOylation, we also tested the effect of the reduction of global SUMOylation in the gut. Knockdown of *Smt3*/*SUMO* in the gut using *NP1-Gal4^{ts}* was again protective under infective conditions (Fig 2D). Further, knockdown of the enzymes that contribute to SUMO conjugation, namely *SAE1*/*Aos1*, *SAE2*/*Uba2*, was also protective, extending adult fly lifespan during *Pe* infection (Fig 2D), as was the expression of a dominant-negative allele of the conjugating enzyme, *Lwr*/*Ubc9* (*Ubc9^{DN}*). Under non-infective conditions, loss of SUMO conjugation appeared to affect lifespan, with *Aos1* knockdown reducing lifespan significantly compared to *NP1-Gal4^{ts}*> *w^{1118}* (Fig 2E).

The protective effect of *Jra* knockdown suggests that the *Pe* may be resistant to clearance by the host-defense response in the presence of *Jra*. We measured the colony-forming units (CFUs) in the gut of adult flies, 2 hours, 8 hours and 24 hours post-infection (hpi, Fig 2F). The number of CFUs gradually decreased as the infection progressed in *w^{1118}* and *Jra^{IA109}*/+. However, there were fewer CFUs in the gut, by order of magnitude, in *Jra^{IA109}*/+ animals, when compared to controls at 8 and 24 hours post-infection, suggesting that absence of Jra signaling in the gut epithelium led to better clearance of gut microbes. Early clearance of the pathogen could also lead to decreased damage of the gut epithelium and a concomitant longer lifespan.

Infection with *Pe* damages the gut epithelium, triggering the ISCs to divide asymmetrically to replace the differentiated cells [46]. Immunostaining with an anti-phospho-Histone 3 (pH3) antibody in the gut post-infection allows us to measure the extent of ISC proliferation. For this, *w^{1118}* and *Jra^{IA109}*/+ flies were fed with *Pe* for 8 hours and dividing cells per midgut were counted (S3B Fig). When compared to UC flies, there was a strong induction of mitoses upon infection. The number of dividing cells was similar between *w^{1118}* (I) and *Jra^{IA109}*/+ (I), suggesting that Jra might not play a dominant role in regulating the intestinal stem cell proliferation upon an immune challenge in the current context.

The reduction of Jra activity during a *Pe* infection appears to be protective to the fly in the context of the gut. Although *Jra* transcripts increase in the gut epithelium in response to *Pe* infection, peaking at 12 hours (next section); their absence reduces the CFUs and also enhances the survival of flies post-infection. Further, reduction of activity of *Bsk* is also protective, suggesting that Bsk and Jra function in a common Jun kinase (JNK) regulatory pathway in the gut. The gut thus appears to merit a detailed study on the roles for SUMO conjugation of Jra. Our data *prima facie* suggests that Jra acts as an important regulator of the host defense response in the gut epithelium.

## Jra is a major regulator of the transcriptional response to Pe infection

To understand the effect of reduced Jra function in the clearance of *Pe* and extending lifespan in infective conditions, we measured the changes in the midgut transcriptome using 3'mRNA sequencing (S4A Fig) and described in Materials and Methods). To start with, we measured the kinetics of activation of *Jra* in the gut during the infection and also the extent of knockdown at each time point using quantitative real-time PCR (qRT-PCR; Fig 3A). We find that *Jra* levels peak at 12 hpi in our experimental setup. In terms of efficiency of knockdown, the *Jra* levels were reduced by at least 60% during the time course of the infection (Fig 3A).

Next, transcriptomes of the midguts were measured at 4, 12, and 24 hpi for both wild-type and after *Jra* knockdown. Principal component analysis (PCA) was performed to estimate the robustness of the dataset (S4B Fig). We plotted the first two principal components and observed that the data sets were distinct as the infection progressed and were seen as separate clusters (S4B Fig). We compared the transcriptional changes in *NP-Gal4^{ts}* >*Jra^{RNAi}* (>*Jra^{RNAi}*

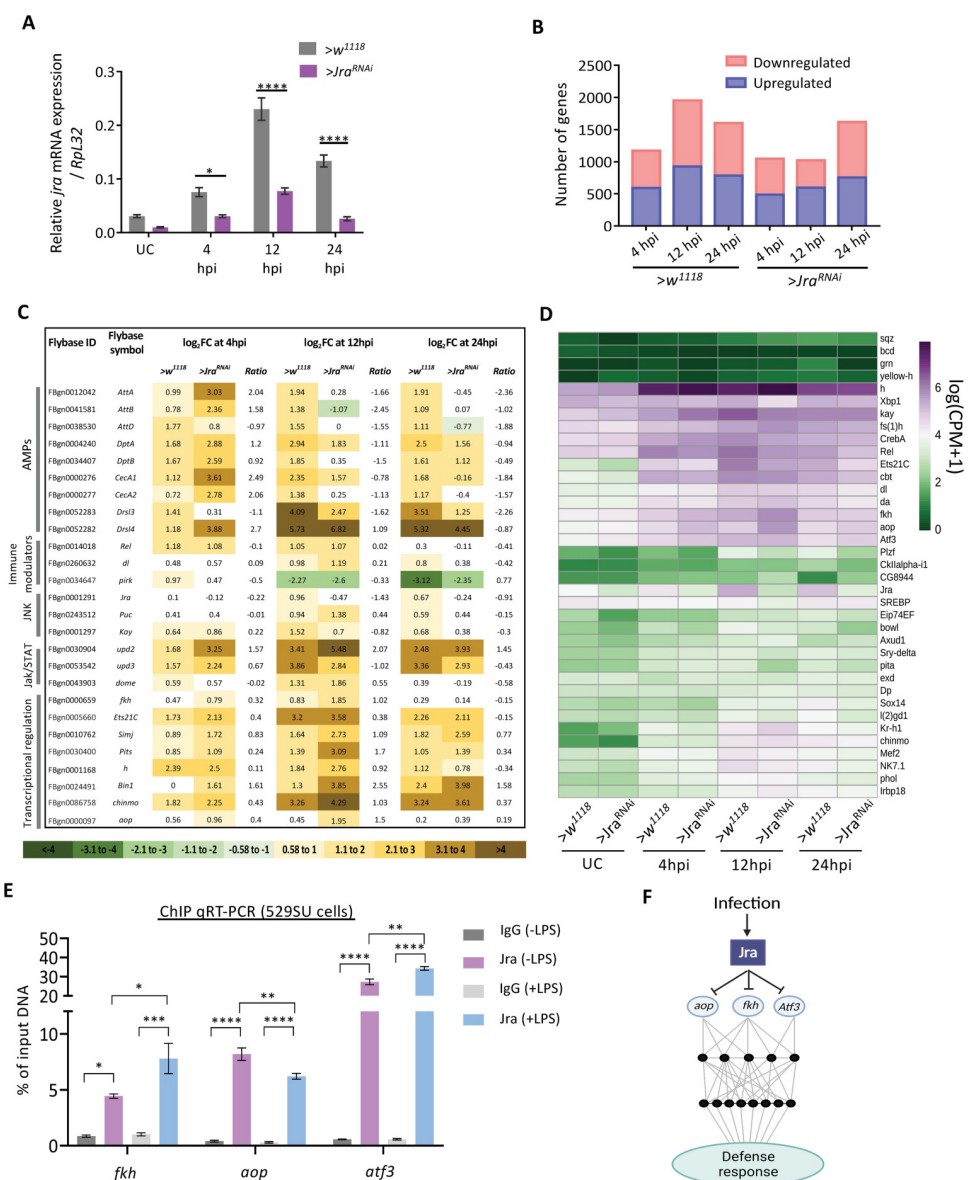

**Fig 3. Jra is a major regulator of the *Pe* infected gut transcriptome. A.** Kinetics of activation of Jra transcripts, as measured by qRT-PCR in $>w^{1118}$ and $>Jra^{RNAi}$ without and with gut infection. *p = 0.0144, ****p<0.0001 as determined by 2-way ANOVA with Bonferroni's post-hoc test for multiple comparisons. Data from three independent experiments. Means and SEMs represented. **B.** Bar plot representation of a total number of significantly differentially expressed genes (FDR<0.1) obtained by comparing $>w^{1118}$ and $>Jra^{RNAi}$ under UC conditions post a 3' mRNA sequencing experiment. **C.** Tabular representation of $\log_2$FC values of transcripts for a few categories of defence genes regulated during the infection. $\log_2$FC values in the range 0.58 to -0.58, indicating no significant changes compared to UC, are marked in white. The $\log_2$FC ratios for each experimental condition are also shown. **D.** Jra binds to the promoters of several genes (data plotted from modERN dataset ENCSR471GSA). Heatmap represents the normalised expression counts of genes for TFs, enriched in the RNA-seq data set, with Jra binding peaks on the promoters. **E.** Binding of Jra to the promoters of *fkh*, *aop* and *atf3* with and without LPS induction in 529SU cells, as quantitated by RT-PCR post a Jra specific Chromatin Immunoprecipitation (ChIP) assay. **F.** Schematic of Jra immune regulatory network representing the effect of Jra on a subset of TFs. ChIP-PCR data and RNA-seq data taken together confirms the repression of *fkh*, *aop* and *atf3* by Jra in regulating the immune response.

here onwards) to *NP-Gal4^{ts}* >*w^{1118}* (>*w^{1118}* here onwards), without an immune challenge (Unchallenged; UC). Upon knockdown, *Jra* is downregulated by 1.2 times on a log 2-fold change (log$_2$FC scale; S1 Table). We also report that 292 genes were differentially expressed with a false discovery rate (FDR) <0.1, amongst which 168 genes were upregulated and 124 genes were downregulated in >*Jra^{RNAi}* as compared to >*w^{1118}* (S4C Fig and S1 Table). The genes including components of immune signalling are putative direct (or indirect) transcriptional targets of Jra. The gut-specific AMPs *Drosomysin-like 2* (*Drsl2*), *Drsl3* and peptidoglycan recognition proteins *PGRP-SC1a* and *PGRP-SC1b* are upregulated (S4D Fig), suggesting a role for Jra in regulating defense genes even in the absence of infection.

Next, we compared the transcriptome of >*w^{1118}* and >*Jra^{RNAi}* in response to infection (I). We normalised data from each infection time point to UC of that genotype and have separated the total genes that were differentially expressed in each pairwise comparison. We then looked at the number of genes differentially expressed at each timepoint in >*w^{1118}* and >*Jra^{RNAi}*. The genes identified ranged from 1000–2000 at each time point. Of note, at 12 hpi, the number of significantly different genes in >*Jra^{RNAi}*, were numerically half of that in >*w^{1118}*, indicating a major perturbation in the immune transcriptome (Fig 3B). We also compared the genes that were common to both the genotypes and genes that were unique to either of the genotypes during infection (FDR<0.1; S5A Fig). A total of 3505 genes were differentially expressed in both the genotypes, with all timepoints taken together (S1 Table). Gene ontology (GO) analysis on all the 3505 genes indicates enrichment of several key GO terms like metabolic process, response to stress, Toll and Imd signalling pathway (S5B Fig).

Fig 3C categorizes a representative set of genes. Analysing log$_2$FC values of individual genes (S1 Table) at the three kinetic time points allows us to understand quantitative differences between the immune response for the two genotypes. At the level of the gene transcripts, the effect appears to be context-specific; most AMPs (Fig 3C) are upregulated post-infection. In >*Jra^{RNAi}* genes such as *AttA*, *AttB*, *DptA*, *DptB*, *CecA1*, *CecA2*, *Drsl4* show higher log$_2$FC values and appear to be activated strongly in the earliest time point, i.e. 4 hpi, further confirmed by qRT-PCR (S6A and S6B Fig) for *DiptA* and *AttD*. Log$_2$FC values for the Imd pathway members, *Relish* (*Rel*), seem to be comparable between both the genotypes.

The same is true for the NFκB transcription factor (TF) *dorsal* (*dl*) and members of the JNK pathway, *Puckered* (*Puc*) and *Kay*. However, *poor Imd response upon knock-in* (*Pirk*), the negative regulator of the Imd pathway [47], had higher log$_2$FC values in >*Jra^{RNAi}* compared to *w^{1118}* at 12hpi. Also, the Jak/STAT pathway components *Upd2* and *Dome* show higher log$_2$FC values in >*Jra^{RNAi}* suggesting that the Jak/STAT pathway might be strongly activated in the guts of >*Jra^{RNAi}* flies. Interestingly, transcriptional regulators like *forkhead* (*fkh*), *Ets at 21C* (*Ets21C*), *chronologically inappropriate morphogenesis* (*chinmo*), *simganj* (*simj*), *Protein interacting with Ttk69 and Sin3A* (*Pits*), *hairy* (*h*), *Bicoid interacting protein 1*(*Bin1*), and *anterior open* (*aop*) show stronger levels of activation in >*Jra^{RNAi}* throughout the infection. This suggests that Jra could be a major player in a larger gut-specific i-GRN that is activated during an immune response.

Jra can influence the transcription of genes by directly binding to their promoters. We hypothesised that Jra could directly influence the transcription of a subset of the 3505 genes obtained in our experiment. To test this, we used the Jra ChIP-seq dataset available from the modERN consortium [48] and mapped the occupancy of Jra on the promoters of 3505 genes (S7A Fig and S1 Table). We identified a total of 778 genes with enriched peaks of Jra binding on the promoters (S7A Fig) and considered them as direct transcriptional targets of Jra. We plotted the normalized counts of these 778 genes across all timepoint and two genotypes and observed clear differences in the expression of several genes in >*Jra^{RNAi}* as compared to >*w^{1118}* (S7B Fig). We then performed GO analysis on this subset and confirmed that 'gene-

specific transcription' is one of the most represented GO terms (S7C Fig). Interestingly, there were 37 TFs in this list, with each TF having the ability to upregulate/downregulate its own set of target genes. Thus, the GRN of Jra, along with its target TFs, would define the core Jra-dependent immune gene regulatory network (the Jra i-GRN). Hence, we had a closer look at the expression patterns of these TFs (Fig 3D). Of the 37 TFs, a subset of them responds strongly to infection. These include *h*, *kay*, *Cyclic-AMP response element-binding protein A* (*CrebA*), *Rel*, *Ets21C*, *cabut* (*cbt*), *dorsal* (*dl*), *daughterless* (*da*), *fkh*, *aop*, *Activating transcription factor 3* (*Atf3*), *Kruppel homolog 1* (*Kr-h1*), *chinmo*, *Myocyte enhancer factor 2* (*Mef2*), *NK7.1*, and *Jra*. Reduction in *Jra* affected *Rel*, *Ets21c*, *Promyelocytic leukaemia zinc finger* (*Plzf*), and *brother of odd with entrails limited (bowl)* consistently in all the time points and can be seen as reduced levels of transcripts suggesting positive regulation by Jra. Genes like *h*, *fkh*, *aop*, *atf3* and *CG8944* show increased transcript levels upon knockdown of Jra, suggesting negative regulation by Jra. In order to validate our targets, we performed a Chromatin Immunoprecipitation (ChIP) assay, using in 529SU cell extracts, on the promoters of a few putative target genes using a rabbit polyclonal antibody, generated against the full-length Jra. We first validated the specificity of the antibody to bind Jra *in-vitro* and *in-vivo* (S8 Fig). We then treated the cells with crude LPS for 3 hours to evoke a robust immune response. We first looked at the enrichment of the antibody over IgG and observed that all the promoter regions amplified show significant enrichment (S9 and 3E Figs), suggesting that *Rel*, *fkh*, *Atf3*, and *aop* are true transcriptional targets of Jra. The promoters of *Jra* and *Kay* were also amplified as AP-1 factors are known to regulate their own transcription. We also amplified promoter sequences of *Puc* and *Mmp1*, which are known transcriptional targets of Jra as positive controls (S9 Fig). Importantly, we observed that the promoters of *fkh*, *Atf3* and *aop* show a change in occupancy of Jra upon LPS treatment (Fig 3E). The ChIP data thus supports our hypothesis, that Jra can regulate transcription of the numerous genes we have identified in our RNA sequencing experiment, by binding directly to their promoters.

Our data from the transcriptome analysis suggests that loss of Jra in the gut both enhances and suppresses a set of genes. The kinetics of upregulation of the AMPs in $>Jra^{RNAi}$ appears to be faster than $>w^{1118}$, and this could lead to early clearance of bacteria and thus increase the longevity of the infected animal. The Jra i-GRN appears to contain several key transcriptional factors with known gut-specific roles in maintaining homeostasis. These, among several others, include *Rel*, *fkh*, *aop*, and *Atf3*, which are also confident targets of Jra. Quantitative ChIP-PCR analysis reveals that the occupancy of endogenous Jra on the promoters of *fkh*, *aop* and *Atf3* changes post immune stimulus regulating the transcription of these targets. The influence of Jra on the transcription of these factors could be part of a larger network that is required to activate and suppress genes to evoke a robust immune response (Fig 3F). Physiologically, the integrated result of the reduction of Jra function and subsequent kinetic modulation of the i-GRN appears to strengthen the host defences, to delay the final victory of the pathogen.

## Generation of Jra<sup>SCR</sup> by CRISPR/Cas9 genome editing

Having established the functional landscape of Jra function in the gut, we focused on understanding specific roles for SUMO conjugation of Jra. For this, we generated a Jra SCR animal using CRISPR/Cas9 technology. This technology has emerged as a powerful tool to edit the genome and generate point mutations within a gene [49,50]. A major benefit of the methodology is the directed nature of the process and the absence of modifications distal from the targeted locus, with the expression of the mutant allele driven by the native promoter. Our strategy involved the use of the Scarless system [50], where a mutant *Jra* genomic template was

available after double-stranded break by Cas9 for homology dependant repair (HDR). The stages for generating a $Jra^{K29R+K190R}$ were as follows. A guide RNA (gRNA) that targeted the third exon in the *Jra* locus was designed and cloned into the pBFv-U6.2B vector (Fig 4A). A transgenic line was generated by targeting the integration of this vector into the second chromosomal attP40 site by embryonic microinjection (Fig 4A). The single guide-RNA line ($Jra^{sg2}$) was balanced and validated. The two SCR mutations that we planned to incorporate into the genomic locus of *Jra* were cloned in the pHD-Scarless-DsRed vector ([50]; Fig 4B). In the Scarless vector, the DsRed cassette is flanked by PBac inverted repeats (IR) that target a TTAA site, and the IRs, in turn, are flanked by the homology regions (HR) specific to the genomic region of interest (Fig 4B). The DsRed is driven specifically in the eye by the P3 promoter and serves as a proxy for the insertion of the entire cassette into the genomic locus. We incorporated the desired K->R mutations in the 3'HR region and assembled two fragments of *Jra* and two vector fragments using Gibson assembly creating the pHD-Scarless-$Jra^{SCR}$ vector (Fig 4B), which was validated by DNA sequencing.

pHD-Scarless-$Jra^{SCR}$ was injected into 600 embryos expressing Cas9 and $Jra^{sg2}$ (Fig 4C and 4D). The F1 animals that emerged were balanced with a 2$^{nd}$ chromosome balancer. Lines that were successfully balanced were screened for positive insertions of the pHD-Scarless-$Jra^{SCR}$ by the presence of red fluorescence in the eye of the fly. HDR seemed to be efficient, with 82 lines showing integration. A small subset (twenty) of DsRed positive lines were crossed with a transposase line. The intent was to excise the DsRed cassette. The TTAA site that is duplicated upon insertion of the cassette is restored into a single genomic TTAA site leaving no 'scar'. Six out of the 20 lines showed a loss of DsRed fluorescence, and these flies were rebalanced and sequenced. Only two out of the six lines had K->R mutations at both positions (29$^{th}$ and 190$^{th}$), and we have extensively characterised these lines in the further sections. Three lines harboured K->R mutation only on the 29$^{th}$ position and not the 190$^{th}$ position. One line did not harbour any K->R mutations, and this line was used as a CRISPR control for all the further experiments and referred to as $Jra^{WT}$ (Fig 4E). The *Jra* genomic region was sequenced to confirm that no new mutations were generated in the region. All the six lines were homozygous viable, isogenic and lost their balancers in a couple of generations.

## SUMO conjugation of Jra attenuates the suppression of immune response by Jra

Earlier, we have shown that Jra is a major regulator of the gut i-GRN. Mechanistically, Jra directly or indirectly regulates key immune effector genes in the gut of *Drosophila* during infection. A comparison of the gut immune transcriptome between $Jra^{WT}$ and $Jra^{SCR}$ animals would lead to an understanding of the role of SUMOylation in regulating Jra function (Next Section). Before that, we looked for phenotypical differences in $Jra^{WT}$ and $Jra^{SCR}$ during an infection to attribute an allelic definition to $Jra^{SCR}$. $Jra^{SCR}$ flies fed with *Pe* responded poorly to infection and succumbed early (Fig 5A). This was true for both lines (L1, L2) that we generated. Interestingly, in terms of survival against infection, $Jra^{SCR}$ flies were more sensitive, unlike the *Jra* hypomorph (compare Figs 2A and 5A). In addition to this, we also performed the survival assay on $Jra^{K29R}$ fly lines (L1 and L2) we generated. In principle, Jra in these flies can still be SUMOylated on K190. Interestingly, $Jra^{K29R}$ flies also show decreased lifespan upon *Pe* infection. However, this phenotype is weaker as compared to $Jra^{SCR}$ flies (S10B Fig), suggesting that the strength of the phenotype is related to the degree of SUMOylation of Jra. To define the nature of the SCR allele, we crossed the $Jra^{WT}$, and $Jra^{SCR}$ flies to $Jra^{IA109}$ and performed lifespan experiments on the heterozygotes. Post feeding with *Pe*, $Jra^{IA109}/Jra^{WT}$ flies lived longer than homozygous $Jra^{WT}$ flies upon infection, consistent with the idea that loss of

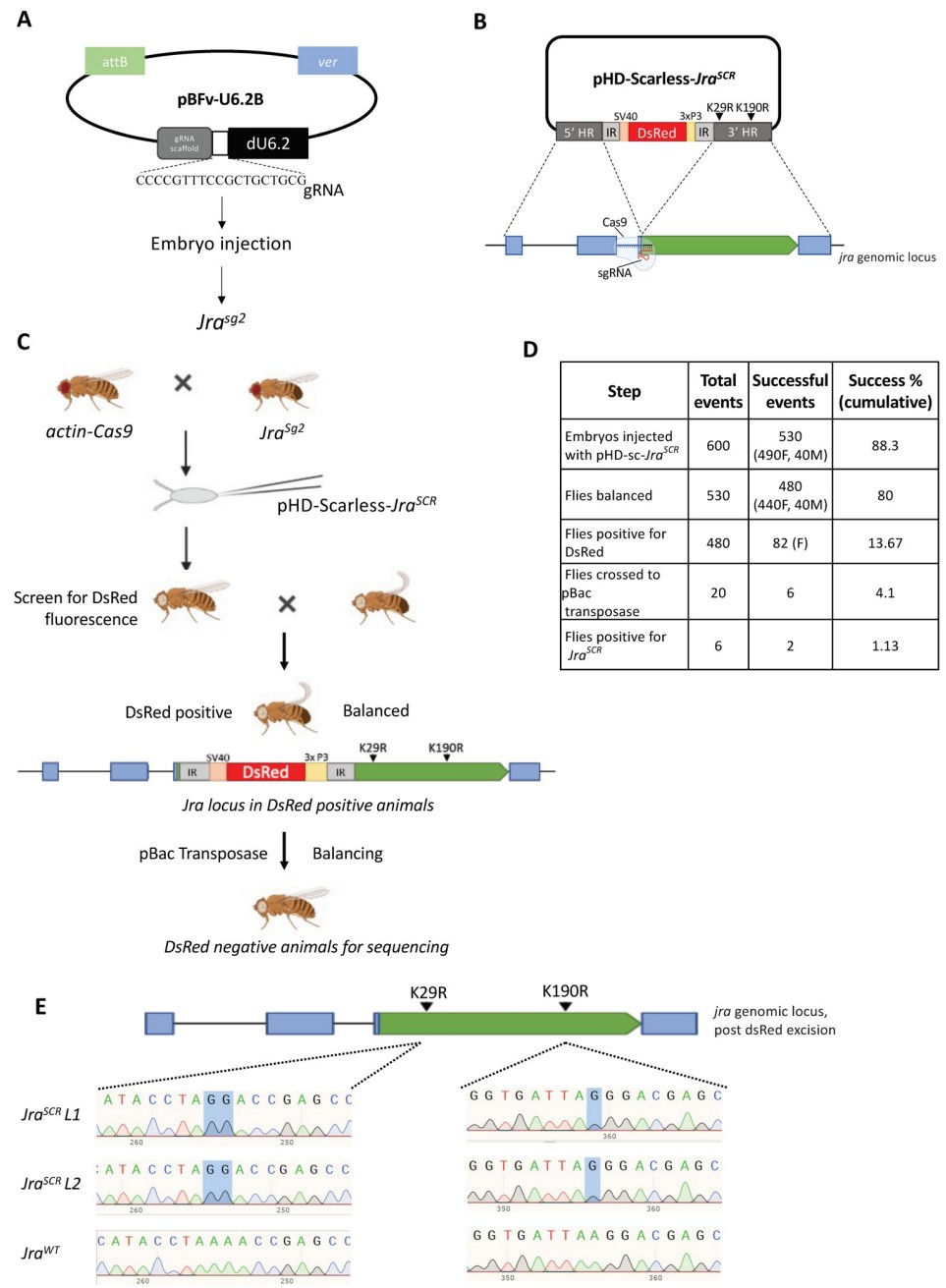

**Fig 4. CRISPR/Cas9 genome editing is used to generate the $Jra^{SCR}$ line. A.** The guide RNA transgenic line, $Jra^{sg2}$, was created for generating a double-stranded break in the $Jra$ locus. A $Jra$ locus-specific gRNA fragment (5'CCCCGTTTCCGCTGCTGCG3') was designed and inserted into the pBFV-U6.2B vector. **B.** Design and cloning strategy for generating the pHD-scarless-$Jra^{SCR}$ vector. Homology regions (HR, in dark grey) from the $Jra$ locus (bottom panel) were cloned into the 5' and 3' arms of the vector as shown in the Schematic. The majority of Jra coding region was part of the 3' arm. As part of the design, the gRNA (panel A) was targeted to the CDS which was a part of the 3rd exon. Details of cloning of the vector are listed in Materials and Methods. The construction of the pHD-Scarless-$Jra^{SCR}$ was validated by sequencing. **C.** Schematic of injection of pHD-Scarless-$Jra^{SCR}$ and genetic crosses implemented to mutate the genomic locus of $jra$ and generate a $Jra^{SCR}$ transgenic fly using CRISPR/Cas9 technology. *actin-Cas9* flies were crossed with the stable Jra gRNA transgenic fly, $Jra^{sg2}$. The embryos from the cross were injected with pHD-Scarless-$Jra^{SCR}$. Flies that emerged were screened for the presence of fluorescent red eyes. DsRed positive flies, with the pHD-Scarless-$Jra^{SCR}$ cassette integrated into the Jra locus, were balanced against the w-; Cyo balancer. SCR mutations would presumably have been incorporated as part of the 3' HR arm. Next, excision of the dsRed cassette was carried out by crossing the flies to a line expressing pBac transposase. Lines in the next generation were screened for loss of DsRed. The lines were balanced, post excision and $Jra$ locus sequenced to confirm the loss of

DsRed and gain of SCR mutations (panel E). *Images for this figure have been generated using BioRender.* **D.** Tabular representation of the efficiency of CRISPR/Cas9 mediated genome editing of Jra. 600 embryos of the Cas9 and gRNA cross were injected with pHD-Scarless-*Jra^SCR*. 530 embryos emerged into adults, of which 490 were females, and 40 were males. Only 82 of 490 females had red fluorescence in their eyes. 20 fly lines out of the 82 positives were randomly selected and crossed to flies expressing PBac transposase. 6 flies showed loss of red fluorescence in the eye, and all were sequenced. Only 2 of the 6 lines harboured both the desired lysine to arginine mutations. **E.** Snapshot of the chromatogram of sequences showing K29R and K190R mutations in the genomic locus of Jra. L1 and L2 represent two independent lines of *Jra^SCR*, and a third line that did not harbour either of the mutations, was used as wildtype control.

*Jra* was protective, enhancing the ability of the fly to fight the infection. *Jra^IA109*/*Jra^SCR* animals also survived longer when compared to homozygous *Jra^SCR* flies, but not when compared to *Jra^IA109*/Jra^WT.

To further characterize the allele, we generated overexpression (OE) lines for both *Jra^WT* and *Jra^SCR*. Wild-type and SCR sequences were cloned into the pUASp-attB vector and UAS-

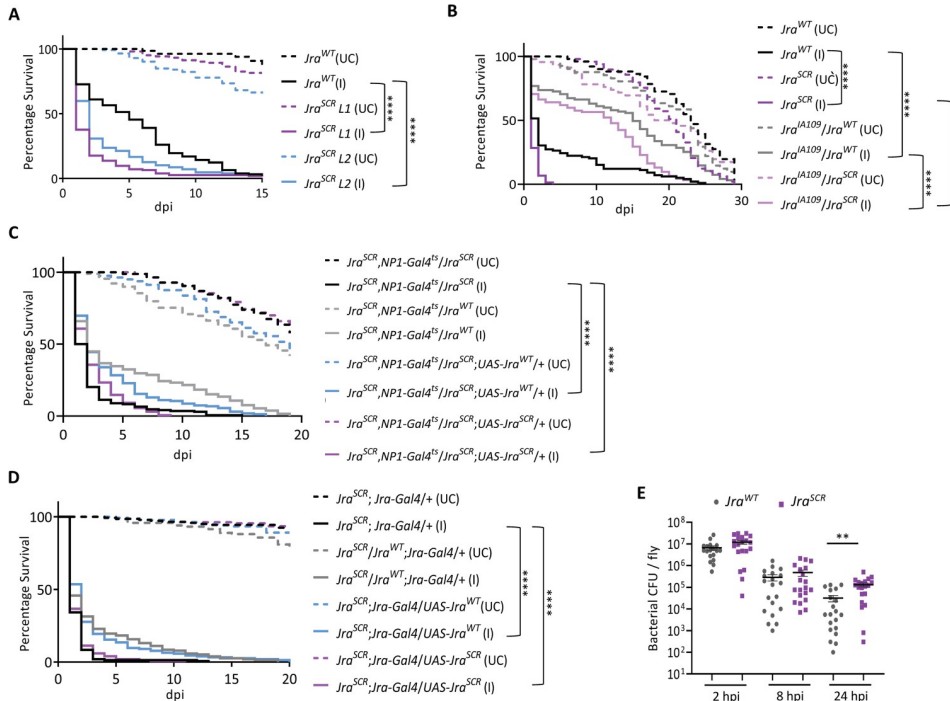

**Fig 5. Survival curves post-infection suggest that *Jra^SCR* is a hypermorphic allele. A.** Survival curves of unchallenged (UC, dashed lines) and orally infected (I, closed lines), CRISPR/Cas9 modified *Jra^SCR* flies, compared to *Jra^WT*. L1 (red) and L2 (blue) indicate two independent *Jra^SCR* lines used in the experiment. Log-rank test for trend was used to individually compare *Jra^SCR* L1 (I) and *Jra^SCR* L2 (I) to *Jra^WT* (I). ****p<0.0001. **B.** Survival curves of UC (dotted) and I (closed), flies: *Jra^WT* (black), *Jra^SCR* (purple), *Jra^IA109*/*Jra^WT* (grey) and *Jra^IA109*/*Jra^SCR* (lavender). Log rank test for trend was used to compare *Jra^IA109*/*Jra^WT* (I) to *Jra^WT* (I), *Jra^IA109*/*Jra^SCR* (I) to *Jra^SCR* (I) and *Jra^IA109*/*Jra^SCR* (I) to *Jra^IA109*/*Jra^WT* (I). ****p<0.0001. **C.** Survival curves of UC (dotted) and I (closed) flies: *Jra^SCR*,*NP1-Gal4^ts*/ *Jra^SCR* (black), *Jra^SCR*,*NP1-Gal4^ts*/*Jra^WT* (grey), *Jra^SCR*,*NP1-Gal4^ts*/*Jra^SCR*;*UAS-Jra^WT*/+ (blue) and *Jra^SCR*,*NP1-Gal4^ts*/ *Jra^SCR*;*UAS-Jra^SCR*/+ (purple). Log rank test for trend was used to compare *Jra^SCR*,*NP1-Gal4^ts*/*Jra^WT* (I), *Jra^SCR*, *NP1-Gal4^ts*/*Jra^SCR*;*UAS-Jra^WT*/+ (I) and *Jra^SCR*,*NP1-Gal4^ts*/*Jra^SCR*;*UAS-Jra^SCR*/+ to *Jra^SCR*,*NP1-Gal4^ts*/*Jra^SCR* (I). ****p<0.0001. **D.** Survival curves of UC (dotted) and I (closed) flies: *Jra^SCR*;*Jra-Gal4*/+ (black), *Jra^SCR*/*Jra^WT*;*Jra-Gal4*/ + (grey), *Jra^SCR*;*Jra-Gal4*/*UAS-Jra^WT* (blue) and *Jra^SCR*;*Jra-Gal4*/*UAS-Jra^SCR* (purple). Log rank test for trend was used to compare *Jra^SCR*/*Jra^WT*;*Jra-Gal4*/+ (I), *Jra^SCR*;*Jra-Gal4*/*UAS-Jra^WT* (I) and *Jra^SCR*;*Jra-Gal4*/*UAS-Jra^SCR* to *Jra^SCR*;*Jra-Gal4*/+ (I). ****p<0.0001. **E.** Scatter dot plot representing bacterial load as colony-forming units (CFUs) post oral feeding with *Pe* Student's t-test was performed comparing *Jra^WT* and *Jra^SCR* at respective time point post-infection. **p = 0.0015. *Data related to the number of animals per experiment and replicates listed in S7A Fig. Data not significant is not represented.*

$Jra^{WT}$ and UAS-$Jra^{SCR}$ lines balanced on the third chromosome. Flies with one copy of $Jra^{WT}$ and $Jra^{SCR}$ in a heterozygous combination survived better than homozygous $Jra^{SCR}$ flies suggesting that a single copy of WT Jra is sufficient to rescue the lethality and restore homeostasis in the gut (Fig 5C). Also, OE of $Jra^{WT}$ in the gut was sufficient to partially rescue the survival seen in $Jra^{SCR}$ homozygous flies, while overexpression of Jra$^{SCR}$ in the gut of homozygous $Jra^{SCR}$ flies did not show any change in lifespan post-infection (Fig 5C). Since the rescue with OE of $Jra^{WT}$ in the enterocytes was only partial, we assumed that precise expression of Jra was required to completely rescue the early lethality seen in homozygous $Jra^{SCR}$ flies. For this, we used Jra specific *Gal4* (*Jra-Gal4*) and looked at the rescue of $Jra^{SCR}$ with OE of $Jra^{WT}$. Similar to what was previously observed, flies with a single copy of $Jra^{WT}$ and $Jra^{SCR}$ in a heterozygous combination survived better than homozygous $Jra^{SCR}$ flies. Interestingly, as hypothesised, OE of $Jra^{WT}$ with *Jra-Gal4* completely rescued the survival of homozygous $Jra^{SCR}$. OE of $Jra^{SCR}$, however, did not alter the survival of the homozygous $Jra^{SCR}$ flies. This suggests that the expression of Jra indeed needs to be tightly regulated to evoke a robust immune response and maintain homeostasis. Also, the role of other organs/tissues in contributing to the expression of Jra cannot be ruled out as *NP1-Gal4$^{ts}$* is specific to the adult enterocytes. Our data rule out $Jra^{SCR}$ as a loss of function allele or a dominant-negative allele. Based on the behaviour of the allele and the equivalent data from hypomorphic alleles (Fig 2A and 2B), $Jra^{SCR}$ appears to be a hypermorphic allele with increased activity of Jra. This would also suggest that SUMO conjugation of Jra could reduce Jra activity, an observation previously reported in the mammalian ortholog, c-Jun [28,29].

Next, we tested if the sensitivity of $Jra^{SCR}$ flies correlates with the ability of the pathogen to survive in the gut. For this, we performed the bacterial clearance assay on $Jra^{WT}$ and $Jra^{SCR}$ flies. We observed that $Jra^{SCR}$ flies were weaker in their ability to clear *Pe* as there were more CFUs as compared to $Jra^{WT}$ at 24 hpi (Fig 5E). This suggested that the defence factors produced by the gut were weaker in Jra$^{SCR}$ animals. We also tested for the ability of ISCs to proliferate post-infection in $Jra^{WT}$ and $Jra^{SCR}$ flies. In both cases, pH3$^{+}$ cells increased, indicating a stem cell response, but the response appears to be equal in both genotypes (S10C Fig). SUMOylation of Jra does not appear to have a role in regulating the division of ISCs during infection.

We further looked to see if $Jra^{SCR}$ was different from $Jra^{WT}$ in terms of subcellular localization. To test this, we probed for endogenous Jra in the guts of $Jra^{WT}$ and $Jra^{SCR}$ flies with and without *Pe* infection. We observed that in both the genotypes, Jra was abundantly found in the nuclei (S11A Fig). Upon oral infection with *Pe*, there was an increase in fluorescence, indicating an increase in Jra protein levels. However, there was no change in localization of Jra between both the genotypes after infection (S11A Fig). We performed a similar experiment in 529SU cells transiently expressing 6XHis tagged Jra$^{WT}$ and Jra$^{SCR}$, as independent experiments. In both genotypes, we observed that Jra was localised to the nucleus and nuclear periphery. Also, there was no significant change in localization of Jra between His-Jra$^{WT}$ and His-Jra$^{SCR}$ post LPS induction. Genetic experiments in this study established $Jra^{SCR}$ as an active allele of Jra (S11B Fig). One possible way Jra$^{SCR}$ could be more active is if it was more stable. To check for stability, we performed a Cycloheximide (CHX) pulse-chase experiment in 529SU cells transiently transfected with 6XHis tagged Jra$^{WT}$ and Jra$^{SCR}$, independently. In both variants, we observed a gradual decrease in protein levels with an increase in the time of CHX treatment (S11C Fig). We quantitated the relative levels of His-Jra and observed the same (S11D Fig). However, we did not observe any significant change in the stability of Jra when comparing His-Jra$^{WT}$ to His-Jra$^{SCR}$. In our experiments, SUMOylation does not appear to influence protein stability as dramatically as it appears to do so as in the context of acetylation and phosphorylation [28–30].

The function of $Jra^{SCR}$ is thus distinct from $Jra^{WT}$. Compared to Jra hypomorphs, the effect is contrary, both in terms of lifespan and bacterial clearance. This further agrees with our assertion that $Jra^{SCR}$ is a hypermorphic allele. Also, since we did not observe and significant differences in terms of localization and stability between $Jra^{WT}$ and $Jra^{SCR}$, we propose that loss of SUMOylation of Jra increases the activity of Jra and thus reduces the ability of the fly to fight infection.

## $Jra^{SCR}$ suppresses the activation of defence genes

To gain further mechanistic insight into the role of $Jra^{SCR}$, we have performed a comparative 3'mRNA sequencing experiment with the guts isolated from $Jra^{SCR}$ and $Jra^{WT}$ at 4 and 12 hours post *Pe* infection. PCA indicates that the samples of a particular genotype form distinct clusters (S12B Fig) with the progression of infection. *Jra* levels in $Jra^{WT}$ and $Jra^{SCR}$ flies were comparable during infection, indicating similar levels of activation (Figs 6A and S12A). Comparing $Jra^{WT}$ and $Jra^{SCR}$ in UC animals, we find that the transcriptional differences were restricted to a very small number (87; FDR<0.1; S2 Table) of genes (Fig 6B). Genes that are known to regulate immune response were found to be both upregulated, (e.g., *Drosomycin-like 2* (*Drsl2*), *Attacin-D* (*AttD*) and *Charon*) and downregulated, (e.g. *Peptidoglycan recognition protein -SC1a* (*PGRP-SC1a*), *PGRP-SC1b* and *Pirk*), in the $Jra^{SCR}$ flies (Fig 6C). Post *Pe* infection, we identified a total of 3134 (FDR<0.1; S2 Table) genes to be differentially expressed at 4 hpi and 12 hpi, as normalised to UC in both the genotypes taken together. These genes are plotted in terms of number upregulated downregulated at each time point of the respective genotype (Fig 6D). We also observed that the changes in genes that were differentially expressed were maximum at 12 hpi (Figs 6D and S12C), an observation we previously made (Fig 3) suggesting that major changes in gene regulation occur at 12 hpi in our experimental setup. Gene ontology analysis reveals several key GO terms like metabolic process, gene expression, response to stress, MAPK signalling pathway etc., to be significantly enriched (S12E Fig) in the genes that are differentially expressed. In order to relate this data to earlier data on Jra loss of function, we evaluated the same set of genes as in Fig 3E. The log$_2$FC values of a majority of AMPs (*AttB*, *AttD*, *Cecropin A2* (*CecA2*), *Diptericin A* (*DptA*), *Drsl3*, and *Drsl4*) were lower in $Jra^{SCR}$ as compared to that of $Jra^{WT}$. The lower levels of AMPs could be one of the reasons for $Jra^{SCR}$ flies to succumb early upon infection. Interestingly, the log$_2$FC values of the TF *Rel* were considerably lower in $Jra^{SCR}$ than $Jra^{WT}$ in both the infection time points. Also, the log$_2$FC values of the negative regulator (of the Imd pathway), *Pirk*, appeared to be higher in $Jra^{SCR}$ flies than $Jra^{WT}$ flies indicating that in $Jra^{SCR}$ flies, Imd signalling may be dampened due to higher Pirk levels. Since Rel is a master immune regulator, we validated this result by performing a quantitative real-time PCR (qRT-PCR) on the guts of $Jra^{WT}$ and $Jra^{SCR}$ post-infection.

We found that the activation of *Rel* was suppressed in the guts of $Jra^{SCR}$ (S14A Fig), the same was true with the activation of *DptA* (S14B Fig) and *AttD* (S14C Fig), which are transcriptionally activated by Rel. In contrast, the log$_2$FC values of components of the JNK pathways were comparable in $Jra^{WT}$ and $Jra^{SCR}$. A number of transcriptional regulators that showed higher log$_2$FC values in >$Jra^{RNAi}$ (Fig 3C) in comparison to $w^{1118}$ showed lower log$_2$FC values in $Jra^{SCR}$ when compared to $Jra^{WT}$. Also, the levels of *Upd3* and *Dome* were lower in the $Jra^{SCR}$ flies. Taken together, this suggests that at a molecular level, the genes necessary to negate *Pe* infection are either suppressed or insufficiently activated in $Jra^{SCR}$, leading to the better survival of the pathogen.

As before (Fig 3), we focused on the genes that could be directly regulated by Jra. For this, we mapped the occupancy of Jra on the 3134 differentially expressed genes. We identified 702

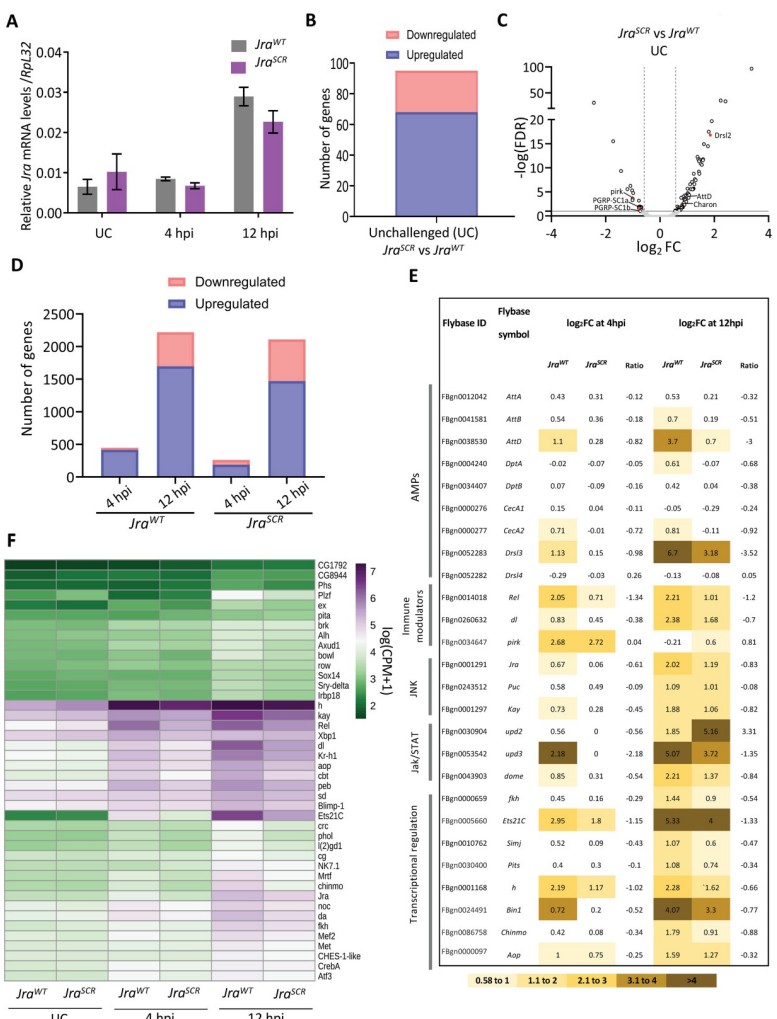

**Fig 6. The modulation of the immune response by Jra^SCR is distinct from that of Jra^WT. A.** Kinetics of activation of Jra transcripts, as measured by qRT-PCR in *Jra^WT* and *Jra^SCR* without and with gut infection. Data from three independent experiments. Means and SEMs represented. **B.** Bar plot representation of a total number of significantly differentially expressed genes (FDR<0.1) obtained by comparing *Jra^WT* and *Jra^SCR* under UC conditions post a 3' mRNA sequencing experiment. **C.** Volcano plot showing the up or down-regulated genes, on comparing *Jra^WT* and *Jra^SCR*. A few genes previously known to function in regulating immunity are highlighted. **D.** Bar plot representation of the total number of significantly differentially expressed genes (FDR<0.1) obtained at 4 and 12 hpi after comparing to UC animals for the respective genotype. **E.** Tabular representation of log₂FC values of transcripts for a few categories of defence genes regulated during the infection, with a color code at the bottom of the table. Log₂FC values in the range 0.58 to -0.58, indicating no significant changes compared to UC, are marked in white. **F.** Heatmap representing the normalised expression counts of genes for TFs, enriched in the RNA-seq data set, with Jra binding peaks on their promoters.

genes with enriched Jra binding peaks on the promoters (S13A Fig). Plotting the normalised expression of the 702 genes indicated changes between both the genotypes as the infection progressed (S13B Fig and S2 Table), and as previously observed in >*w^1118* and >*Jra^RNAi* dataset, the 'gene-specific transcription' GO term was one of the most represented categories (S13C Fig). We evaluated gene function and identified 42 TFs in the list. We plotted the normalised expression counts between the two genotypes during infection to look for changes in the expression pattern of these TFs (Fig 6F). Surprisingly, we observed that the mRNA levels in *Jra^SCR* animals for this subset of genes were lower as compared to *Jra^WT* as the infection

progressed. This decrease was considerable in *kay*, *Rel*, *dl*, *aop*, *Kr-h1 cbt* and *Ets21C* at 12 hpi (Fig 6F) and modest for *h*, *pebbled* (*peb*), *chinmo*, *Jra*, *da*, *fkh* and *Atf3*.

The early lethality seen in *Jra^{SCR}* flies thus strongly correlates with the findings from the transcriptomics data. We observed that in *Jra^{SCR}*, several AMPs and key genes that regulate the immune response were insufficiently activated, allowing the pathogen to thrive at the expense of the host. The Jra i-GRN previously described appears to be weakly activated in the *Jra^{SCR}* flies shifting the balance in favor of the pathogen. The mRNA levels of *Rel*, in particular, are significantly lowered in *Jra^{SCR}* (S14 Fig), suggesting that SUMOylation of Jra is an important event for robust activation of *Rel* in the gut (S14D Fig).

## Discussion

### JNK and Rel signalling interact to regulate immune signalling

The Jun-kinase (JNK) pathway is a well-characterized signalling module used in a wide variety of cellular processes, including, but not limited to proliferation, immunity, apoptosis, and embryonic development. The module is complex with multiple receptors, signalling interme-diates, and transcriptional effectors (reviewed by [51–55]). A central theme of the pathway is a cascade of kinases that lead to the activation of a number of the effector TF(s), including c-Jun. In *Drosophila*, the components of the JNK pathway have been well studied [18,22,24,56–60] with Jra also implicated as a transcriptional effector of the systemic immune response in *Dro-sophila* [61,62](S15 Fig). Current models for the role of Jra in immune signalling for gram-neg-ative infections place it downstream of the PGRP-LC receptor. The pathway bifurcates downstream of Immune Deficient (Imd), with the MAP kinase kinase kinase Tak1 signalling to both Jra and Rel (S15 Fig). The pathway, which is well characterized in flies, involves the IKK complex Kenny/Ird5 and leads to the phosphorylation and subsequent cleavage of Rel. The N-terminal 68 kD fragment of Rel then translocates to the nucleus where it activates the host-defense response. In a parallel path, Tak1 signals to the Jun kinase-kinase Hep, which sig-nals to the JNK Bsk, which in turn phosphorylates and activates Jra (S15 Fig). Jra can function to negatively regulate genes of the host immune response, with a specific example of the AMP *AttA*, where Jra competes with Rel, and shuts off *AttA* transcription [61,62]. Both Jra and Rel are thus effectors of immune signalling [63–68] and there may be considerable overlap between the gene-regulatory networks they activate.

### Attenuation of Jra signalling favors the host

Our interest in the functional consequences of SUMO conjugation in the immune response led us to Jra, which is a SUMO target. Also, the role of Jra has not been studied in the context of the gut immune response, making it an exciting candidate to explore. As a first step, we used loss of function Jra alleles, gut-specific Jra knockdowns, and infection with *Pe* to elucidate the role of Jra in regulating the gut immune response. Knockdown of either *Jra* or *Bsk* leads to an increase in lifespan post-infection, concomitant with a decrease of the *Pe* population in the gut, suggesting that JNK signalling is active in the gut and that pathogen infectivity is facilitated by increased JNK/Jra signalling. Additionally, reduction of global SUMO conjugation in gut ECs was also protective, suggesting that the availability of the SUMOylation machinery during infection was helpful to *Pe*. Seminal papers in the field have shown that pathogens such as *Lis-teria* [69], *Shigella* [70] and *Salmonella* [71] capture the mammalian host SUMO conjugation machinery to improve their infectivity. *Pe* may also use similar molecular mechanisms as a means to modulate host SUMOylation. We have previously reported that several proteins in signalling pathways undergo a change in SUMOylation status upon an immune challenge

[12], and it would be extremely interesting to study the context-specific effect of SUMOylation on these proteins during an immune response.

The immune transcriptome of the gut, defined as the set of the genes activated or repressed on *Pe* infection was significantly altered on *Jra* knockdown. Of note were AMPs, *AttA*, *AttB*, *DptA*, *DptB*, *CecA1*, *CecA2*, and *Drsl4*, with comparatively higher transcript levels for the critical early time-point of 4 hpi and *Pirk*, an important negative regulator of Imd signalling with lower transcript levels at 4 and 12 hpi. Transcriptional regulators (Fig 3C) were mostly upregulated. *fkh* [72,73], *dl*, *Chinmo* [74], *Ets21C* [75,76], *atf3* [42,77], and *Aop* [78,79] are examples of TFs whose transcripts are upregulated at 4 and 12hpi. In addition to this, the JAK/STAT components, Upd2 (at 4hpi, 12hpi and 24hpi), Upd3 (12hpi) and Dome (12hpi) are elevated in Jra knockdown. ChIP experiments suggest that Jra binds to the promotor sequences of *Rel*, *Fkh*, *aop*, *Atf3* and *Kay* and regulates their transcription. *fkh*, *aop* and *Atf3* are representative of genes where Jra acts as a repressor or co-repressor as knockdown of *Jra* leads to an increase in their transcription, with ChIP data confirming that Jra binds directly to their promoters. Interestingly, the occupancy of Jra is altered upon an immune stimulus (Fig 3F), for the promoters of these genes.

## SUMO conjugation of Jra also favors the host

The *Jra^{SCR}* flies, generated by the CRISPR/Cas9 mediated genome editing, are sensitive to infection, when compared to *Jra^{WT}* and succumb early. One contributory factor to this sensitivity is the persistence of bacteria in the *Jra^{SCR}* flies, suggesting a weaker host-defense response to the pathogen. In contrast, the loss of Jra function, validated using different hypomorphic alleles, led to a longer lifespan and comparatively fewer bacteria under infective conditions. This would indicate that the lack of Jra SUMOylation favors the pathogen and that ability of Jra to get SUMO conjugated is detrimental to infectivity. Intriguingly, reduction of *Jra* in the gut and subsequent reduction of Jra signalling favors the host. This suggests that the Jra^{SCR} allele evokes a less than robust immune response. An analysis of the differential gut transcriptional response to the pathogen in *Jra^{SCR}* and *Jra^{WT}* animals suggests that the differences lie in the ability of the wild type allele to differentially regulate a subset of defense genes. Important TFs in the immune GRN that are direct targets of Jra are insufficiently activated as compared to controls and these include *Rel*, *dl*, *fkh*, *Ets21c*, *chinmo* and *aop*. In strong contrast to Jra knockdown, these genes are downregulated in *Jra^{SCR}*, when compared to controls in both 4 and 12 hpi. *Pirk* is upregulated, especially at 12 hpi and the kinetics of transcriptional activation of AMPs such as *AttA*, *AttB*, *DptA*, *DptB*, *CecA1*, *CecA2*, and *Drsl3* is slower, as underscored by the negative ratio's at 4 hpi (Fig 6E). The insufficient activation of *Rel* in *Jra^{SCR}* flies is especially notable as Rel is the major transcriptional regulator of the immune response, and lowered production of Rel would directly influence Imd signalling and subsequent activation of defense genes. Perturbation of secreted factors such as *upd2* and *upd3* could also influence inter-organ signalling [80,81], indicating possible long-range, systemic effects for Jra and its SUMOylation. We hypothesize that the lowered activation or repression of key TFs would influence the transcription of the immune GRN and compromise the fly's ability to fight infection (Fig 7A). We therefore propose that SUMOylation of Jra is a crucial step to fine-tune the immune response and maintain homeostasis in the gut of *Drosophila*.

## An integrated model for the effect of SUMO conjugation on Jra

How does SUMOylation reduce the activity of Jra in the context of transcriptional regulation? Traditionally, Bsk phosphorylates Jra on the N-terminus, leading to the activation of Jra [16,21], while the mode of deactivation of Jra has been elusive to date. We propose

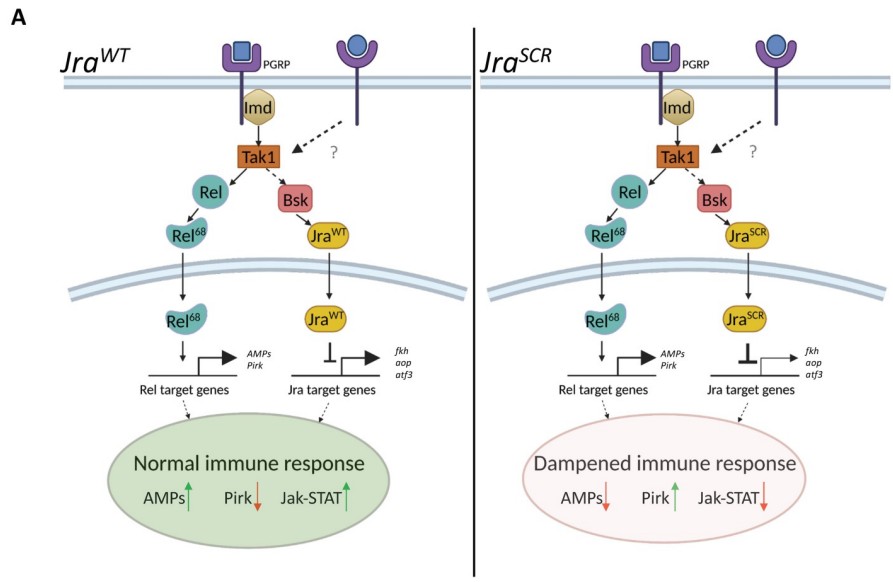

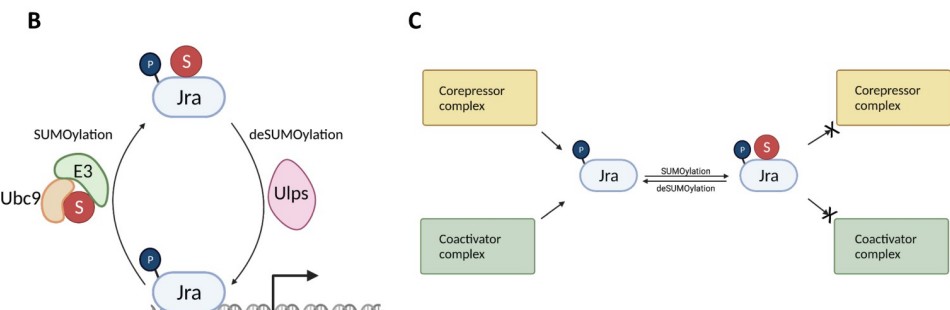

**Fig 7. SUMOylation of Jra regulates the gut immune response. A.** Peptidoglycan recognizing receptors on the cellular surface recognize pathogens and activate the Imd signalling cascade. Both Rel and Jra are transcriptional effectors for this pathway and they act to evoke a robust immune response. Our data places Jra as an important effector and modulator of the immune GRN (left panel). In a scenario where Jra is resistant to SUMO conjugation (right panel), we find that the immune response is dampened, favouring the proliferation of the pathogen. **B.** A small fraction of SUMOylated Jra can influence the robustness of the host defense response in the gut. We propose that the SUMO conjugation/deconjugation of Jra is a dynamic process, regulated by the nuclear SUMO conjugation machinery, and occurring on the promoters Jra target genes. Dynamic cycles of SUMOylation fine-tunes the activity of Jra with SUMOylated Jra maintained at low levels, *i.e.* <<5% of total Jra. **C.** The mechanism for effect of SUMO conjugation on Jra mediated transcription in *Drosophila* is not clear. Multiple possibilities exist; SUMOylation may influence activation and/or repression by Jra, and this activity may depend on context, with SUMOylated Jra interacting differentially with the transcriptional machinery at Jra target promoters. *Images for this figure have been generated using BioRender.*

SUMOylation of Jra as a mechanism to attenuate the function of Jra in the context of the gut immune response. We hypothesise that SUMO conjugation/de-conjugation is a dynamic event that happens near the promoters of Jra target genes, and the two states are in equilibrium (Fig 7A and 7B). Depending on the context, there is a shift in this equilibrium, which dictates the outcome of Jra activity. During infection, the transcript levels of Jra rise, leading to an increase in protein levels of Jra. More Jra protein would lead to more active Jra, which could be detrimental to the host and beneficial to the pathogen. Hence, Jra is SUMOylated to reduce its activity and to evoke a potent immune response.

One possibility is that SUMOylation of Jra causes Jra to exit the promoter leading to a decrease in overall transcription activity (Fig 7B). This argument is further strengthened by

data from the mammalian ortholog, c-Jun, where a SUMO conjugation resistant variant shows higher transcriptional activity [28,29]. Consistent with this, the c-Jun binding partner, c-Fos, also shows increased transcriptional activity upon abrogating SUMO conjugation [29]. The SUMOylated form of c-Fos is less abundant on the promoters of select target genes, where SUMOylation knocks c-Fos off the chromatin to disrupt its activity [82]. As Jun and Fos are binding partners and perform similar functions, we strongly believe that the same could be happening with SUMOylation of Jra (Fig 7B). Also, there is evidence that SUMO conjugation negatively regulates the function of a set of TFs [33,83–85]. Another possible mechanism could be a change in interactors, particularly the transcriptional apparatus, that could alter the outcome of Jra activity upon SUMOylation (Fig 7C). However, we do not rule out the possibility of both these mechanisms acting parallelly in a highly context-specific manner on individual genes. The levels of SUMOylation of any protein at a given time is very low ($\ll$5%). To test out either of the proposed hypotheses in the current context, one needs to capture the SUMOylated species in abundant amounts in the gut epithelium. With current tools, it becomes challenging to perform ChIP experiments or mass-spectrometry based assays to enrich the SUMOylated species and understand the context-specific role of SUMO conjugation of Jra.

In summary, our study establishes Jra as an important regulator of the gut immune response. Jra and Rel GRN's have substantial overlap with Jra activity regulating defense genes activated by Rel. SUMOylation of Jra is an important regulatory event in the gut, in the host response to pathogen. SUMO conjugation of Jra is beneficial to the host and detrimental to the pathogen. A pathogen which can reduce or block SUMO conjugation of Jra would gain a better foothold and improve its chances of colonizing the gut.

## Materials and methods

### Fly husbandry

Flies were reared on standard cornmeal agar at 25˚C in a 12-hour light-dark cycle. Only 6–8 day old females were used in all the experiments in this study. All $NP1$-$Gal4^{ts}$ crosses were maintained at 21˚C to inactivate the Gal4. 3 days post eclosion, flies were shifted to 29˚C for 3 days (to activate the Gal4) until bacterial feeding. The flies were shifted to 29˚C from the beginning of the infection till the end of the experiment. Table 1 lists the lines used in our studies.

### Cloning of gRNA and $Jra^{SCR}$ for CRISPR/Cas9 mediated genome editing

The gRNA was designed using http://targetfinder.flycrispr.neuro.brown.edu/. The gRNA sequence with no off-targets and optimal for our experimental design was chosen and cloned into the pBFv-U6.2B vector. The 2.5 kb genomic locus of *jra* was amplified from a single $w^{1118}$ fly. This was used to generate 3'HR and 5'HR fragments of *jra*. The SCR mutations were incorporated in the 3'HR region. The pHD-scarless-DsRed vector was amplified as two separate fragments using specific primers. The two fragments of *jra* along with the two fragments of the vector were ligated using Gibson assembly (NEB) and sequenced. The 3'HR region cloned into the vector was resistant to the gRNA due to the modification of a few degenerate nucleotides and the PAM sequence. For screening of $Jra^{SCR}$ after the excision of DsRed cassette, the genomic DNA was extracted by homogenising single flies in 50ul of a buffer containing 10mM TRIS, 1mM EDTA and 20uM Proteinase K. The homogenate was incubated at 37˚C for 30 min followed by 85˚C for 5 min. 2.5kb of the genomic region of Jra was amplified and sequenced for each line. The $Jra^{SCR}$ and $Jra^{WT}$ fly lines were regularly sequenced to confirm the desired genotype. All the primers used are listed in S3 Table.

**Table 1. Genotypes and source of *Drosophila* lines used for our experiments.**

| Line/Genotype | Source | Description |
|---|---|---|
| $Jra^{WT}$ | This study | Jra locus modified using CRISPR/Cas9. No mutations incorporated into the genome |
| $Jra^{SCR}$ L1 | This study | Jra locus modified using CRISPR/Cas9. K29R+K190R mutations are incorporated into the genome. L1 = line 1. Extensively used in this study |
| $Jra^{SCR}$ L2 | This study | Jra locus modified using CRISPR/Cas9. K29R+K190R mutations are incorporated into the genome. L2 = line 2. |
| $Jra^{K29R}$ L1 | This study | Jra locus modified using CRISPR/Cas9. K29R mutation is incorporated into the genome. L1 = line 1. |
| $Jra^{K29R}$ L2 | This study | Jra locus modified using CRISPR/Cas9. K29R mutation is incorporated into the genome. L2 = line 2. |
| $Jra^{SCR}$; Jra-Gal4 | This study | $Jra^{SCR}$ balanced with Jra-Gal4 |
| $Jra^{SCR}$,NP1-Gal4$^{ts}$/CyO | This study | $Jra^{SCR}$ recombined with NP1-Gal4$^{ts}$ |
| UAS-Jra$^{WT}$ | This study | pUASp-attB-$Jra^{WT}$ inserted into attp40 site |
| UAS-Jra$^{SCR}$ | This study | pUASp-attB-$Jra^{SCR}$ inserted into attp40 site |
| $Jra^{sg2}$ | This study | gRNA targeting exon 3 of jra genomic locus |
| Myo31DF-Gal4, UAS-GFP,tub-gal80$^{ts}$ (NP1-Gal4$^{ts}$) | Sveta Chakrabarti [46] | Enterocyte specific Gal4 line (NP1-Gal4$^{ts}$) |
| y v[1]; P{y[+t7.7] v[+t1.8] = TRiP.JF01184}attP2 | BDSC:31595 | UAS-Jra$^{RNAi}$ |
| y[1] w[1118]; P{w[+mC] = Jbz}10 | BDSC:7218 | UAS-Jra$^{DN}$ |
| w[1118] P{w[+mC] = UAS-bsk.DN}2 | BDSC:6409 | UAS-Bsk$^{DN}$ |
| w[1118]; P{w[+mC] = UAS-Fra.Fbz}5 | BDSC:7214 | UAS-Kay$^{DN}$ |
| cn[1] Jra[IA109] bw[1] speck[1]/CyO | BDSC:3273 | Jra null |
| Adh[fn23] pr[1] cn[1] Jra[76–19]/CyO | BDSC:9880 | Jra null |
| y[1] v[1]; P{y[+t7.7] v[+t1.8] = TRiP.HM05055} attP2/TM3, Sb[1] | BDSC:28569 | UAS-Uba2$^{RNAi}$ |
| y[1] v[1]; P{y[+t7.7] v[+t1.8] = TRiP.HM05183} attP2/TM3, Sb[1] | BDSC:28972 | UAS-Aos1$^{RNAi}$ |
| y[1] sc[*] v[1] sev[21]; P{y[+t7.7] v[+t1.8] = TRiP. HMS01540}attP2 | BDSC:36125 | UAS-SUMO$^{RNAi}$ |
| UAS-lwr$^{DN}$ | Shubha Govind [86] | UAS-Ubc9$^{DN}$ |
| w1118; P{GMR61B05-GAL4}attP2 | BDSC:46459 | Jra-Gal4 |
| y[1] M{Act5C-Cas9.P.RFP-}ZH-2A w[1118] DNAlig4[169] | BDSC:58492 | Actin5c Cas9 |
| w[1118] | BDSC:3605 | $w^{1118}$ |

## Cloning and generation of constructs for overexpression

Jra (FBpp0087498) was amplified from the *Drosophila* gold collection library (https://www.fruitfly.org/EST/gold_collection.shtml) using specific primers (S3 Table). These amplicons were independently cloned into the pGEX-4T1 vector for bacterial SUMOylation assay and pRM-HA3 vector for transfection into S2 cells using a modified Seamless Ligation Cloning Extract (SLiCE) protocol [87]. The site-directed mutagenesis approach with specific primers was used to modify all the lysine residues to arginine residues to abrogate SUMO conjugation. pUASp-AttB vector was procured from the *Drosophila* Genomics Resource Centre (DGRC, #1358) and was used for targeted insertions of WT/SCR constructs into the attP2 site on the 3$^{rd}$ chromosome. All the clones were confirmed by sequencing and used for downstream experiments.

## Bacterial SUMOylation assay

This is a modified in-vitro SUMOylation assay that was previously described [25]. The quartet vector comprising of the *Drosophila* SUMO machinery components was co-transformed with GST tagged Jra. The bacterial culture was induced with 1mM of Isopropyl β-D-1-thiogalacto-pyranoside (IPTG) for 6 hours at 25°C. 10ml of the bacterial culture was harvested in 1ml 50mM *tris* aminomethane (TRIS) buffer containing 150mM NaCl, 1mM Dithiothreitol (DTT), 1ug/ml lysozyme, and 1mM phenylmethylsulfonyl fluoride (PMSF). The cells were lysed using a VibraCell probe sonicator with 2/3sec ON/OFF cycle for 2 min at 60% amplitude.

## S2 cell culture, transfections, cycloheximide treatment, LPS induction and immunostaining

S2 cells that were stably transfected with Flag-SUMO (referred to as 529SU cells) were a kind gift from Prof. Albert Courey. The cells were grown and maintained in Gibco Schneider's *Drosophila* Medium (Thermo Fischer Scientific, #21720024) supplemented with 10% heat-inactivated fetal bovine serum (FBS) (Thermo Fischer Scientific, #10082147) at 23°C. 1ug of plasmid was transfected per 1ml of cells using TransIT-Insect Transfection Reagent (Mirus, #6100) as per the manufacturer's protocol. The cells were induced with 0.5M CuSO4 and harvested after 48h of induction. For the Cycloheximide (CHX) pulse-chase assay, 529SU cells were seeded appropriately, transfected, and induced as described above. Post 48h of induction, the media was removed, and fresh media was added. CHX was added at a final concentration of 60μg/ml. Cells were harvested at regular intervals, and lysates were run on a WB to score for the stability of Jra. Crude LPS (Sigma-Aldrich, #L2630), which contains peptidoglycan contaminants [65] was used to evoke an widespread immune response in 529SU cells at a final concentration of 10 μg/ml. For immunostaining, 529SU cells were grown on a coverslip in a 12 well plate. The cells were fixed with 4% formaldehyde for 20 min and washed with 1X PBS. The cells were blocked with blocking buffer (1X PBS, 0.1% TritonX-100 and 2%BSA) for an hour and incubated with primary antibody diluted in blocking buffer at 4°C overnight. 1:500 of anti-His antibody (SCBT, #sc-8036) was used as primary antibody and anti-mouse alexa-568 (Thermo Fisher Scientific, #A-11004) was used as a secondary antibody. DAPI was used to stain the nuclei.

## Pulldown, Immunoprecipitation and Western blotting

Glutathione affinity PD for the bacterial SUMOylation assay was performed using Pierce Glutathione Agarose beads (Thermo Fischer Scientific, #16101). Post lysis, the bacterial lysate was incubated with the equilibrated beads for 12-14h at 4°C. The beads were then washed 3 times with the lysis buffer with 0.1% Triton-X100, and bound protein was eluted using 1X laemmli buffer. Ni-NTA superflow resin (Qiagen, #30430) was used to pull down His-Jra from 529SU cells as per the manufacturer's protocol. To maintain stringent denaturing conditions, 8M Urea was used throughout the experiment in all the buffers. 20mM N-ethylmaleimide was also used to inactivate the SENPs. Pierce BCA Protein Assay Kit (Thermo Fischer Scientific, #23225) was used to estimate the protein concentration before performing the Western blotting. 1-2mg of total protein for PD and IP experiments and 50ug of total protein for inputs. The following antibodies were used as primary antibodies for the WB. Anti-His (SCBT, #sc-8036; used in 1:2000), anti-GST (SCBT, #459; used in 1:5000) and anti-Flag (Sigma-Aldrich, #F7425; used in 1:2000). Peroxidase conjugated anti-mouse (Jackson Immunoresearch, #115-035-003; used in 1:10000) and peroxidase-conjugated anti-rabbit (Jackson Immunoresearch,

#111-035-003; used in 1:10000) were used as secondary antibodies. All the blots were developed using Immobilon Western Chemiluminescent HRP Substrate (Sigma-Aldrich, #WBKLS).

### Bacterial culture, gut infection, fly survival assays and bacterial clearance assays

*Pseudomonas entomophila* (*Pe*) was used in this study to evoke a robust immune response in the fly gut. The pathogen was always selected for rifampicin (100ug/ml) resistance and hydrolysis of casein on a milk agar plate before the infection experiments. The bacterial pellet obtained from an overnight grown culture was resuspended in a 5% sucrose solution, so the final $OD_{600}$ was ~200. For all the infection experiments, flies were first starved for 2 hours without food and water. Post starvation, the flies were transferred to a fly vial (at 20–30 flies/vial) containing Whatman filter disk that was dipped in the concentrated bacterial solution and the flies were allowed to feed for a certain amount of time specific to the experiment. The flies were transferred to a vial containing standard food post-feeding, and the dead flies were counted every 24 hours. Bacterial feeding was carried out for 24 hours for all the survival experiments except when Jra$^{SCR}$ flies were used, where feeding time was 6 hours. For assessing the bacterial load, flies were fed on *Pe* for 2 hours and transferred to vials containing standard food. Before plating, the flies were rinsed in 70% ethanol and left to dry. Whole flies were crushed in sterile PBS, and the lysate was plated on a bacterial agar plate containing rifampicin. The bacterial plates were imaged, and colonies were counted in Fiji (ImageJ) post an intensity cutoff for each plate.

### Gut dissections, immunostaining, microscopy western blotting

The guts were precleared by growing the flies on 5% sugar agar overnight for all the immunostaining experiments. The next day, flies were fed with *Pe* for 8 hours after 2 hours of starvation. Post feeding, the guts were dissected in ice-cold 1X PBS and fixed in a solution containing 1X PBS, 0.1% TritonX100 and 4% paraformaldehyde for 20 minutes. The guts were then incubated at room temperature for an hour in a blocking solution that contained 1X PBS, 0.5% bovine serum albumin, 0.3% TritonX100 and 3% normal goat serum. This was followed by incubation with respective primary antibodies specific to the experiment; 1:200 anti-phospho-Histone3 (pH3) antibody (Cell Signalling Technology, #9701L) diluted in blocking solution and incubated with tissue at 4˚C overnight; 1:500 of purified anti-Jra (inhouse) diluted in blocking solution and incubated with tissue at 4˚C overnight. Anti-rabbit alexa568 (Thermo Fisher Scientific, #A-11011) was used as a secondary antibody in 1:1000 dilution. DAPI was used to mark the nuclei, and the samples were mounted in Slowfade gold antifade mountant (Thermo Fisher Scientific, #36940). The mounted samples were imaged using a 20X and 40X oil immersion objective on a Leica SP8 confocal microscope. pH3 positive cells per gut were counted using ImageJ after specifying a threshold and size cutoff (4–20μm$^2$). Anti-Jra antibody (SCBT, #sc398615) was used in 1:1000 to probe for Jra in the guts post WB in S12A Fig.

### qRT-PCR

1ug of extracted RNA was used to generate cDNA using a high-capacity cDNA reverse transcriptase kit (Thermo Fisher Scientific, #4374966) using the manufacturer's protocol. The cDNA was diluted in a 1:5 ratio for all the experiments. qRT-PCR was performed using gene-specific primers (S3 Table) with Kapa SyBr Fast (Sigma-Aldrich, #KK4618) in an Applied Biosystem ViiA 7 real-time PCR system.

## Jra antibody generation

Full length Jra was cloned into pET-45b(+) vector with an N-terminal 6XHis tag and was expressed in *E.coli* BL21(DE3) strain. The expressed protein was purified using Ni-NTA super-flow resin (Qiagen, #30430). The purified protein was run on an Amersham S200 preparative column as a second step of purification. Purified full length Jra was used to immunize a New Zealand white rabbit at Bioklone, India. Upon attaining appropriate titers, the serum was used to purify Jra specific antibodies by cross adsorption to protein A/G agarose. Purified Jra antibody was tested in WB and IFC for specificity (S8 Fig).

## Chromatin Immunoprecipitation (ChIP) assay

529SU cells were grown in $75cm^2$ corning flasks with 15 ml culture volume. Crude LPS was added to a final concentration of 10μg/ml. Cells were fixed using 1% methanol free formaldehyde. The fixation reaction was quenched with 150 mM Glycine. The fixed cells were washed with 1X PBS and resuspended in a swelling Buffer (25 mM Tris-HCl pH 7.9, 1.5 mM $MgCl_2$, 10 mM KCl, 0.1% NP-40, 1 mM DTT, 0.5 mM PMSF, 1X PIC) and homogenised using Dounce homogeniser to lyse the cells and release the nuclei. The nuclei were pelleted, resuspended in sonication buffer (50 mM Tris-HCl pH 7.9, 140 mM NaCl, 1 mM EDTA, 1% Triton X-100, 1% SDS, 0.1% Sodium deoxycholate, 0.5 mM PMSF, 1X PIC), and incubated for 30 min on ice. Nuclei were sonicated to obtain an average chromatin size of 200–700 base pairs using Covaris M220 focused ultrasound sonicator. Immunoprecipitations were carried out in ChIP buffer (16.7 mM Tris-HCl pH 8.0, 167 mM NaCl, 1.2 mM EDTA, 1.1% Triton-X 100, 0.01% SDS, 1X PIC) with 20μg of chromatin using 6μg of purified anti-Jra antibody at 4˚C for 14–16 h. Purified anti-rabbit IgG control was used for every condition. The samples were then incubated with 100 μl of a Surebeads Protein A magnetic beads (Biorad) (that were saturated with tRNA and BSA) for 3h at 4˚C. ChIP samples were reverse-crosslinked, and the DNA was purified using the Phenol:Chloroform:Isoamylalcohol-based precipitation method. Input DNA that was obtained after sonication was also purified accordingly. Purified DNA was subjected to quantitative PCR.

## RNA isolation, 3'mRNA library preparation and sequencing

Total RNA was extracted from 50–60 midguts/experiment using RNeasy Plus Universal Kits (Qiagen, #73404) according to the manufacturer's protocol. 3' mRNA specific libraries from three independent biological replicates were amplified using QuantSeq 3' mRNA-Seq Library Prep Kit FWD (Lexogen, #015.96) using the manufacturer's instructions. The concentrations of the libraries were estimated using Qubit dsDNA HS Assay Kit (Thermo Fisher Scientific, #Q32851). Quality assessment and library size estimation of the individual libraries was done using an HS DNA kit (Agilent, #5067–4626) in a Bioanalyzer 2100. The libraries were pooled in equimolar ratio, and single-end 75bp reads were sequenced on the Illumina HiSeq 2500 platform.

## Read mapping, counts generation and differential expression analysis

On average, 4–5 million reads were uniquely mapped per sample. Sequencing quality was assessed using FastQC v0.11.5. Post QC, the reads were mapped to the *Drosophila* genome (dm6) using STAR aligner v.2.5.2a [88]. Gene expression levels were measured using the counts generated by HTSeq-count v 0.6.0 [89]. To obtain differential expression of the genes, the biological conditions were compared pairwise using DESeq2 [90]. The above steps were carried out on the bluebee platform (https://lexogen.bluebee.com/quantseq).

## ChIP-seq data analysis

Jra specific ChIP-seq dataset (ENCSR471GSA) against an input dataset (ENCSR908EFA) was obtained from the model organism Encyclopedia of Regulatory Networks (modERN) consortium (https://epic.gs.washington.edu/modERN/) [48]. Input normalised Bigwig files were used, and the occupancy of Jra was plotted on the gene body (+/- 2kb of the mRNA) of all the significantly differentially expressed genes from the 2 RNA sequencing datasets using deeptools 3.5.0 [91].

## Bioinformatic analysis

The FDR cutoff < 0.1 was considered for the entire analysis. Normalised counts were generated using the CPM function of edgeR [92]. The PCA was performed using a custom R script. GO enrichment analysis of the significantly differentially expressed gene set was performed using gProfiler (https://biit.cs.ut.ee/gprofiler/gost) [93] and panther database (http://pantherdb.org/). GO:0003700 'DNA-binding transcription factor activity' term was used to fetch the TFs used in the analysis.

## Statistics

All experiments were performed in three biological replicates unless otherwise specified. Adequate sample size was chosen for each experiment and appropriate statistical test was performed using GraphPad Prism 8.0.

## Supporting information

**S1 Fig. GST is not SUMO conjugated.**
(PDF)

**S2 Fig. Expression of JNK components in different cell types of the adult gut.**
(PDF)

**S3 Fig. Jra regulates the gut immune response.**
(PDF)

**S4 Fig. Gut transcriptomics of $>w^{1118}$ and $>Jra^{RNAi}$.**
(PDF)

**S5 Fig. Comparison of Gut transcriptomics of $>w^{1118}$ and $>Jra^{RNAi}$ in terms of overlap and gene ontology.**
(PDF)

**S6 Fig. $>Jra^{RNAi}$ shows increased expression of AMPs during infection.**
(PDF)

**S7 Fig. Mapping Jra occupancy on the promoters of differentially expressed genes in $>w^{1118}$ and $>Jra^{RNAi}$.**
(PDF)

**S8 Fig. Validation of the anti-Jra antibody.**
(PDF)

**S9 Fig. Jra regulates transcription by directly binding to the promoters of key transcription factors.**
(PDF)

**S10 Fig. *Jra^{SCR}* dampens the gut immune response.**
(PDF)

**S11 Fig. *Jra^{WT}* and *Jra^{SCR}* are similar in terms of expression levels and protein stability.**
(PDF)

**S12 Fig. Gut transcriptomics of *Jra^{WT}* and *Jra^{SCR}*.**
(PDF)

**S13 Fig. Mapping Jra occupancy on the promoters of differentially expressed genes in *Jra^{WT}* and *Jra^{SCR}*.**
(PDF)

**S14 Fig. *Jra^{SCR}* shows insufficient activation of key immune effectors.**
(PDF)

**S15 Fig. The Imd/JNK pathway in the *Drosophila* immune response.**
(PDF)

**S1 Table. Differentially expressed genes in the gut, post knockdown of *Jra*.**
(XLSX)

**S2 Table. Differentially expressed genes in the guts of *Jra^{WT}* and *Jra^{SCR}*.**
(XLSX)

**S3 Table. Primers used in this study.**
(XLSX)

## Acknowledgments

We thank: Bloomington *Drosophila* Stock Center (BDSC), Indiana, supported by NIH grant P40OD018537, for fly stocks; Fly facility at the National Centre for Biological Sciences (NCBS), Bangalore for embryonic injections. Dr. Deepti Trivedi for her input on design of the CRISPR Scarless experiments; Dr. Sveta Chakrabarti for providing protocols and reagents related to the infection experiments. Prof. Sanjeev Galande, Dr. Krishnapal Karmodia, Dr. Saurab Pradhan and Dr. Akhila Gungi for their inputs on the NGS experiments, reagents and machine time for experiments. Snehal Patil and Yashwant Pawar for fly media and stock maintenance.

## Author Contributions

**Conceptualization:** Amarendranath Soory, Girish S. Ratnaparkhi.

**Data curation:** Amarendranath Soory, Girish S. Ratnaparkhi.

**Formal analysis:** Amarendranath Soory, Girish S. Ratnaparkhi.

**Funding acquisition:** Girish S. Ratnaparkhi.

**Investigation:** Amarendranath Soory, Girish S. Ratnaparkhi.

**Methodology:** Amarendranath Soory, Girish S. Ratnaparkhi.

**Project administration:** Girish S. Ratnaparkhi.

**Resources:** Girish S. Ratnaparkhi.

**Supervision:** Girish S. Ratnaparkhi.

**Validation:** Amarendranath Soory, Girish S. Ratnaparkhi.

**Visualization:** Amarendranath Soory, Girish S. Ratnaparkhi.

**Writing – original draft:** Amarendranath Soory, Girish S. Ratnaparkhi.

**Writing – review & editing:** Amarendranath Soory, Girish S. Ratnaparkhi.

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
