## [Decision Letter · Decision Letter 0]

11 Oct 2021

Dear Dr Ratnaparkhi,

Thank you very much for submitting your manuscript "SUMOylation of Jun fine-tunes the Drosophila gut immune response" for consideration at PLOS Pathogens. As with all papers reviewed by the journal, your manuscript was reviewed by members of the editorial board and by several independent reviewers. In light of the reviews (below this email), we would like to invite the resubmission of a significantly-revised version that takes into account the reviewers' comments.

We cannot make any decision about publication until we have seen the revised manuscript and your response to the reviewers' comments. Your revised manuscript is also likely to be sent to reviewers for further evaluation.

Sincerely,

Denise M. Monack

Section Editor

PLOS Pathogens

Denise Monack

Section Editor

PLOS Pathogens

Kasturi Haldar

Editor-in-Chief

PLOS Pathogens

orcid.org/0000-0001-5065-158X

Michael Malim

Editor-in-Chief

PLOS Pathogens

orcid.org/0000-0002-7699-2064

Reviewer's Responses to Questions

**Part I - Summary**

Reviewer #1: In this manuscript, Soory and Ratnaparkhi studied the role of the transcription factor Jra in drosophila immune response. Using a large number of mutant flies, they demonstrated that Jra limits the induction of a protective immune response to pathogens. Then they studied the SUMOylation of Jra. First, they identified the main SUMOylated lysines in vitro and in cellulo and second, they analyzed the effect of their mutation on flies’ response to infection and the regulation of gene expression. This manuscript provides interesting new data on the role of SUMOylation in the control of transcription factors activity in a living organism. Nevertheless, except for few genes, the effects on transcription are not always very clear, which questions their biological relevance.

Reviewer #2: In the manuscript entitled “SUMOylation of Jun fine-tunes the Drosophila gut immune response” by Soory and Ratnaparkhi, the authors propose that Jra is a modulator of the immune gene regulatory network induced by Pseudomonas entomophila that is regulated through SUMOylation. They identified K29 and K190 as the SUMOylation sites in Jra. In addition, they carried out transcriptome analysis of a fly line expressing the SUMOylation mutant of Jra and also upon downmodulation of global SUMOylation. Their results indicate that a fly line expressing the Jra SUMOylation mutant is more sensitive than the WT fly line to Pseudomonas infection and suggest that SUMOylation reduces Jra activity. The work has several important problems, including absence of controls, that need to be solved.

Reviewer #3: This manuscript reports how SUMO modification of the transcription factor Jra, the Drosophila Jun ortholog, regulates immune response in the gut upon Pseudomonas entomophila infection. The authors identify K29 and K190 as major SUMO conjugation sites in Jra. Moreover, they show that a decrease in Jra signalling is beneficial in the host response, while expression of the SUMO-deficient JraK29R+K190R mutant is more susceptible to infection with a weakened host response and increased proliferation of Pseudomonas. RNASeq analysis suggests that loss of SUMOylation of Jra alters transcriptional networks of the immune response.

Altogether, the data are of potential general interest in the SUMO field and the experiments are for the most part convincing. One strength of this work is the investigation of Jra SUMOylation in an in vivo model.

**Part II – Major Issues: Key Experiments Required for Acceptance**

Reviewer #1: (No Response)

Reviewer #2: Major concerns:

1. The authors carried out in vitro SUMOylation assays in order to confirm the SUMOylation of Jra and to identify the SUMOylation sites.

This technique has important problems including lack of controls: the authors express Jra proteins fused to GST. GST proteins contain lysine residues that could work as SUMO acceptors. Therefore, the authors must include a GST protein as a negative control in their assays. In addition, to probe that the higher molecular bands detected in the Western-blot using anti-His antibody correspond to Jra SUMOylated proteins they need to demonstrate that these bands are recognized by the anti-GST antibody. Moreover, the authors must show the His-SUMO bands corresponding to both the unconjugated and conjugated His-SUMO protein expressed in the bacteria.

2. The SUMOylation pattern shown in fig 1E differ from the one shown in fig.1D (the highest molecular band is missing and an additional lower molecular band is detected in K233). The SUMOylation analysis of Jra-WT must be also included in this panel.

3. Similar problems are found in the SUMOylation assay in cells. The band corresponding to Jra SUMOylated protein must be detected with anti-Histidine antibody. In addition, the WB with anti-Flag antibody must show both the conjugated and unconjugated SUMO expressed in the cells.

4. Analysis of the SUMOylation of Jra mutants in cells reveals that the Jra-K29R was not SUMOylated, suggesting that mutation of this residue is enough to avoid SUMOylation in cells. Why the results of both SUMOylation assays are different? Some of the fly lines obtained by the authors contain only this mutation. The authors must analyze those fly lines in order to evaluate potential differences with the double mutant.

5. The authors speculate about the protein expression and SUMOylation regulation upon infection but they do not show any experiment demonstrating their suggestions. They must evaluate: i) whether infection modulates Jra SUMOylation in cells and, ideally, in flies; ii) consequences of Jra SUMOylation on protein stability and subcellular localization in cells; iii) analyze whether the changes in mRNA levels at different times after infection occur in parallel to its modulation at protein level and of this with the SUMOylation levels.

Additional questions

Does BSK modulate the SUMOylation of Jra? Does the activation of JNK in response to other stimuli modulate Jra SUMOylation?

English needs to be improved

Reviewer #3: M major concern is that the data are entirely based on the use of the SUMO conjugation resistant JraK29R+K190R mutant. Formally, it cannot be excluded that this mutant lacks other PTMs, such as ubiquitylation or acetylation at the respective lysine residues. This issue could be easily addressed by changing the glutamat residues rather than the lysine residues within the KxE consensus residues around K29 and 190. This would likely prevent SUMOylation, but not other PTMs. Furthermore, the the work falls short in providing mechanistic insight into the SUMO-controlled JRA activity. I do acknowledge that providing such data, e.g. ChIP or interaction proteomics data, might be difficult in an in vivo setting. However, the authors should at least exploit their cell culture system to compare promoter binding of JraK29R+K190R or wild-type Jra on some of the target genes identifies in the in vivo model.

**Part III – Minor Issues: Editorial and Data Presentation Modifications**

Reviewer #1: Figure 3E: it is quite difficult to compare the two conditions W1118 and JRARNAi and have a clear view of the role of JRA in the expression of the genes. It would be easier to have ratio between the two genotypes in addition to the ratios

Figure S6B: here again, the difference between w1118 and JRA-RNAi is not so obvious from the heatmap, although I agree there are some differences. It is important to identify the genes with a Log2FC>1 or <-1 between the two conditions for each time point.

Figure 4: the informations on the generation of JRAscd could be moved to the method part and the figure to the supplementary data.

Figure 5A-B: the conclusion of the role of JRA SUMOylation on flies survival upon pe infection is not clear.

Figure 2 and 5: It is not clear to which comparisons the pvalues are linked to. There should be brackets between the legends on the side of the survival curves

Figure S8: I disagree that the PCA analysis shows that both genotypes form different clusters. The clusters are rather linked to the time of infection than to the genotype.

Global reduction of SUMOylation is protective whilst inhibition of JRA SUMOylation increase sensitibity. The authors should discuss this apparent contradictiona

Reviewer #2: (No Response)

Reviewer #3: (No Response)

PLOS authors have the option to publish the peer review history of their article (what does this mean?). If published, this will include your full peer review and any attached files.

Reviewer #1: No

Reviewer #2: No

Reviewer #3: No
---

## [Decision Letter · Decision Letter 1]

9 Feb 2022

Dear Dr Ratnaparkhi,

We are pleased to inform you that your manuscript 'SUMOylation of Jun fine-tunes the Drosophila gut immune response' has been provisionally accepted for publication in PLOS Pathogens.

Best regards,

Denise M. Monack

Section Editor

PLOS Pathogens

Denise Monack

Section Editor

PLOS Pathogens

Kasturi Haldar

Editor-in-Chief

PLOS Pathogens

orcid.org/0000-0001-5065-158X

Michael Malim

Editor-in-Chief

PLOS Pathogens

orcid.org/0000-0002-7699-2064

Reviewer Comments (if any, and for reference):

Reviewer's Responses to Questions

**Part I - Summary**

Reviewer #1: The authors have successfully adressed all my concerns

Reviewer #2: The authors have addressed this reviewer's concerns and they have improved the manuscript. The authors demonstrate that the modulator of the immune gene regulatory network induced by Pseudomonas entomophily June is regulated by SUMOylation. They identified the SUMOylation sites in Jra and using in vivo studies they revealed that SUMO modulates Jra activity.

**Part II – Major Issues: Key Experiments Required for Acceptance**

Reviewer #1: (No Response)

Reviewer #2: (No Response)

**Part III – Minor Issues: Editorial and Data Presentation Modifications**

Reviewer #1: (No Response)

Reviewer #2: (No Response)

PLOS authors have the option to publish the peer review history of their article (what does this mean?). If published, this will include your full peer review and any attached files.

Reviewer #1: No

Reviewer #2: No

---

## [Editor Report · Acceptance letter]

3 Mar 2022

Dear Prof. Ratnaparkhi,

We are delighted to inform you that your manuscript, "SUMOylation of Jun fine-tunes the Drosophila gut immune response," has been formally accepted for publication in PLOS Pathogens.

Best regards,

Kasturi Haldar

Editor-in-Chief

PLOS Pathogens

orcid.org/0000-0001-5065-158X

Michael Malim

Editor-in-Chief

PLOS Pathogens

orcid.org/0000-0002-7699-2064